# Consistent Supervised-Unsupervised Alignment for Generalized Category Discovery

Jizhou Han[1], Shaokun Wang[2]*, Yuhang He[1], Chenhao Ding[3], Qiang Wang[1], Xinyuan Gao[4], Songlin Dong[5], Yihong Gong[1]*

[1]National Key Laboratory of Human-Machine Hybrid Augmented Intelligence,
National Engineering Research Center for Visual Information and Applications,
Institute of Artificial Intelligence and Robotics, Xi'an Jiaotong University
[2]School of Computer Science and Technology, Harbin Institute of Technology, Shenzhen
[3]School of Software Engineering, Xi'an Jiaotong University    [4]Kuaishou Technology
[5]Faculty of Microelectronics, Shenzhen University of Advanced Technology

{jizhou-han, dch225739, qwang}@stu.xjtu.edu.cn, heyuhang@xjtu.edu.cn,
wangshaokun@hit.edu.cn, dongsl@suat-sz.edu.cn, ygong@mail.xjtu.edu.cn,

## Abstract

Generalized Category Discovery (GCD) focuses on classifying known categories while simultaneously discovering novel categories from unlabeled data. However, previous GCD methods suffer from inconsistent optimization objectives. This inconsistency leads to feature overlap and ultimately hinders performance on novel categories. To address these issues, we propose the Neural Collapse-inspired Generalized Category Discovery (NC-GCD) framework. By pre-assigning and fixing Equiangular Tight Frame (ETF) prototypes, our method ensures an optimal geometric structure and a consistent optimization objective for both known and novel categories. We introduce a Consistent ETF Alignment Loss that unifies supervised and unsupervised ETF alignment while enhancing category separability. Additionally, a Semantic Consistency Matcher (SCM) is designed to maintain stable and consistent label assignments across clustering iterations. Our method achieves state-of-the-art performance on multiple GCD benchmarks, significantly enhancing novel category accuracy and demonstrating its effectiveness.

## 1 Introduction

Generalized Category Discovery (GCD) has raised emerging attention in recent years, which aims to simultaneously classify known categories and discover novel ones from unlabeled data. Unlike traditional semi-supervised learning, which assumes all categories are predefined during training, GCD operates in a more realistic open-world scenario [1, 2, 3] where only a small portion of known-category data is labeled, and all novel-category samples lack both labels and category information. It requires models to utilize limited supervision from labeled known categories while autonomously discovering semantically coherent clusters in the unlabeled subset without assuming predefined category structures or a clear distinction between known and novel distributions.

Recent research in GCD has witnessed substantial progress. Early research [4] introduces a contrastive learning framework to distinguish known and novel categories. On this basis, contrastive learning-based methods [5, 6] refine category boundaries using dynamic contrastive learning and prompt-based affinity learning. Additionally, representation learning-based methods[7, 8, 9, 10] focus on optimizing feature representations to enhance category separability.

---

*Yihong Gong and Shaokun Wang are the corresponding authors.

39th Conference on Neural Information Processing Systems (NeurIPS 2025).

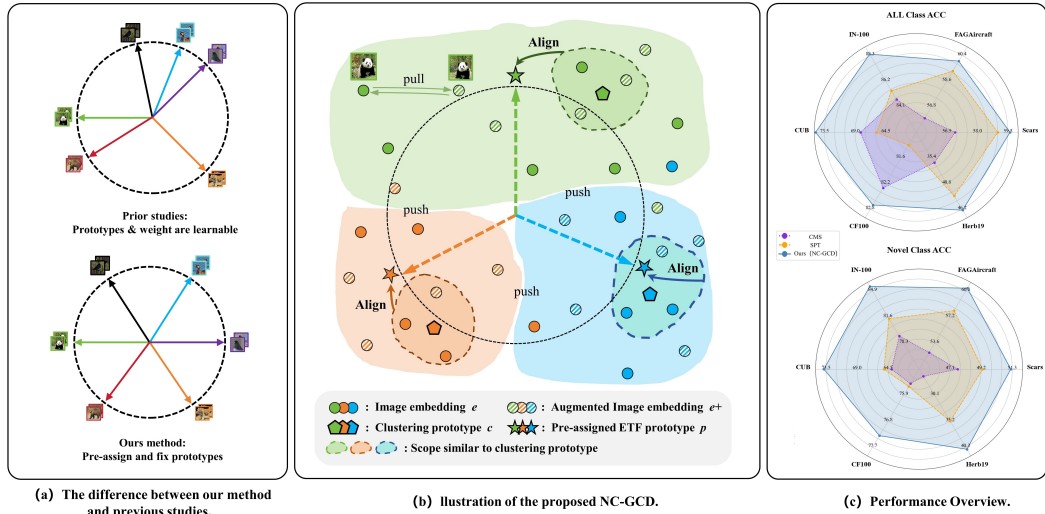

**(a)** The difference between our method and previous studies.

**(b)** llustration of the proposed NC-GCD.

**(c)** Performance Overview.

Figure 1: Illustration of the differences between prior studies, our proposed NC-GCD framework, and Performance Overview. (a) Our method pre-assigns fixed ETF prototypes rather than dynamically learning prototypes, ensuring consistent optimization objectives for known and novel categories. (b) The overview of the NC-GCD. (c) Compared to previous studies, our method exhibits superior overall accuracy, with a notable improvement in the accuracy of novel categories.

However, most existing methods dynamically optimize cluster prototypes or classification weights, leading to two key issues. **First, *Inconsistent Optimization Objective*:** Existing frameworks lack a consistent optimization objective for known and novel categories. As a result, the models tend to prioritize learning from labeled known categories while failing to establish proper decision boundaries for the novel ones. **Second, *Category Confusion*:** As shown in Fig. 1a, existing methods lack geometric constraints on feature distributions. Moreover, novel categories are optimized entirely in an unsupervised manner, making it even more difficult to achieve sufficient separation between feature-similar categories, leading to category overlap and reduced accuracy. Based on the above observations, we wonder:

***Can we pre-assign an optimal geometric structure where both known and novel categories are equally well-separated, enabling consistent learning for all categories by aligning features to this structure?***

Neural Collapse (NC) is a phenomenon in well-trained classification networks, where features of each category align with a Simplex Equiangular Tight Frame (ETF) structure [11]. In this structure, within-category features collapse to their respective category mean, and the means of different categories are at the vertices of a simplex. This alignment maximizes inter-category separation while maintaining within-category compactness, creating an optimal geometric arrangement for classification tasks.

Motivated by Neural Collapse theory, we propose the Neural Collapse-inspired Generalized Category Discovery (NC-GCD) framework, which leverages NC principles to create a structured feature space for both known and novel categories. Unlike existing methods that dynamically learn prototypes, as shown in Fig. 1b, our approach pre-assigns ETF prototypes before training, ensuring an optimal geometric structure and a consistent optimization objective. Our framework integrates three key components: 1) *Unsupervised ETF Alignment* periodically clustering all categories and aligning the top $\alpha\%$ most confident samples to their respective ETF prototypes. This process enhances feature separability, particularly for novel categories. 2) *Supervised ETF Alignment* stabilizes known categories by aligning labeled features with their corresponding ETF prototypes. Afterwards, we unify supervised and unsupervised alignment by designing a *Consistent ETF alignment loss*. However, clustering instability may lead to inconsistent label assignments, where samples of the same category are mapped to different clusters across multiple iterations, thereby disrupting ETF alignment. Even more critically, the direct mapping between ground-truth labels and ETF prototypes in supervised alignment can introduce mismatches, further compromising model stability. 3) To address these issues, we introduce the **Semantic Consistency Matcher** (SCM), which enforces

one-to-one alignment between clusters across iterations, stabilizing pseudo-label assignments and ensuring proper mapping of supervised labels to ETF prototypes. By integrating these components, our method establishes a consistent optimization objective, building a structured feature space that enhances category separability and effectively mitigates category confusion. Our contributions are summarized as follows:

(1) We propose a fixed ETF prototype framework that establishes a consistent optimization objective for both known and novel categories, improving category separability through a consistent supervised-unsupervised alignment.

(2) We introduce the Semantic Consistency Matcher to maintain consistency in cluster labels across multiple iterations, stabilizing the training process and reducing clustering-induced fluctuations.

(3) Our method, as demonstrated in Fig. 1c, achieves strong performance on six GCD benchmarks, with significant improvements in novel category accuracy.

## 2 Related Work

### 2.1 Generalized Category Discovery (GCD)

GCD identifies known and novel categories within a partially labeled dataset. Early methods[4] leveraged contrastive learning to distinguish categories. Later, DCCL [5] and PromptCAL [6] refined category boundaries using dynamic contrastive learning and prompt-based affinity mechanisms. GPC [12] and SimGCD [13] introduced Gaussian Mixture Models and parametric classification frameworks to estimate category distributions. Other methods have focused on self-supervised clustering and feature refinement. PIM [7] and CMS [8] introduced contrastive mean-shift learning and representation alignment strategies to improve novel category separability. Reciprocal learning and class-wise regularization have been introduced in RLCD [14], while ProtoGCD [15] unifies prototype optimization and debiasing for category discovery. Meanwhile, approaches like UNO [16] and ORCA [17] explored semi-supervised representation learning, while RankStats [18] employed ranking-based statistics to facilitate unknown category discovery. Meanwhile, there is a growing line of work on continual generalized category discovery [19, 20, 21, 22]. However, most existing methods rely on dynamically learned prototypes and lack a consistent optimization target. This results in category confusion and imbalanced learning.

### 2.2 Neural Collapse and its Applications

Neural Collapse (NC) describes a geometric phenomenon in deep networks where, in the final stage of training, category feature representations converge to form a Simplex Equiangular Tight Frame structure [11]. This structure ensures optimal category separation and minimal intra-category variance, making it highly effective for classification. Several works have provided theoretical insights into Neural Collapse under different loss functions, demonstrating that it emerges naturally in both cross-entropy loss [23, 24, 25] and mean squared error (MSE) loss [26, 27, 28, 29]. Beyond theoretical exploration, NC has been applied to various machine-learning tasks. In imbalanced learning, NC principles have been leveraged to stabilize classifier prototypes under category-imbalanced training [30, 31]. In few-shot class-incremental learning, studies have explored ETF-based classifiers to enhance feature-classifier alignment and mitigate forgetting [32]. In federated learning, NC has been shown to improve model generalization under non-iid settings [33]. Similar prototype-based mechanisms have also been applied in domain-incremental scenarios, where consistent concept representations are maintained across domains [34]. Furthermore, NC-inspired approaches have been developed to address large-scale classification problems, such as optimizing one-vs-rest margins for generalized NC [35]. TRAILER [36] also draws inspiration from NC; however, it employs a fixed classifier with a cross-entropy–based ETF loss applied to both labeled and pseudo-labeled data, which can cause optimization bias and instability. Our NC-GCD framework extends NC principles by pre-assigning ETF prototypes to known and novel categories. By integrating supervised and unsupervised ETF alignment, our method effectively addresses GCD's misalignment and category confusion.

# 3 Preliminaries

## 3.1 Problem Setting of GCD

Generalized Category Discovery aims to identify novel categories in an unlabeled dataset while maintaining classification performance for known categories. Formally, we define a dataset $\mathcal{D} = \mathcal{D}^l \cup \mathcal{D}^u = \{(x_i, y_i)\}$, where the labeled dataset is given by $\mathcal{D}^l = \{(x_i^l, y_i^l)\} \subset \mathcal{X} \times \mathcal{Y}^l$, containing samples $x_i^l$ with corresponding labels $y_i^l$ from the set of known categories $\mathcal{Y}^l$. The unlabeled dataset is defined as $\mathcal{D}^u = \{x_i^u\} \subset \mathcal{X}$, which consists of samples from both known and novel categories. The overall category set is $\mathcal{Y}^u = \mathcal{Y}^l \cup \mathcal{Y}^n$, where $\mathcal{Y}^l \cap \mathcal{Y}^n = \emptyset$ and $\mathcal{Y}^n$ represents the unknown novel categories. The goal of GCD is to train a model that can accurately classify samples from $\mathcal{Y}^l$ while discovering and organizing samples from $\mathcal{Y}^n$.

## 3.2 Neural Collapse and Definition of Simplex ETF

Neural Collapse, observed during the terminal training phase in well-regularized neural networks on balanced datasets [11], describes a symmetric geometric structure in the last-layer features and classifier weights. Features of the same category collapse to their within-category mean, which aligns with the corresponding classifier weights, forming a Simplex ETF.

**Definition of Simplex ETF.** A Simplex ETF consists of $K$ vectors in $\mathbb{R}^d$ that form an optimal geometric configuration. These vectors, denoted as $P = [p_1, p_2, \ldots, p_K] \in \mathbb{R}^{d \times K}$, satisfy:

$$P = \sqrt{\frac{K}{K-1}} U \left( I_K - \frac{1}{K} \mathbf{1}_K \mathbf{1}_K^\top \right), \tag{1}$$

where $U \in \mathbb{R}^{d \times K}$ is an orthogonal matrix satisfying $U^\top U = I_K$, $I_K$ is the $K \times K$ identity matrix, and $\mathbf{1}_K$ is a $K$-dimensional all-ones vector. Each prototype $p_k$ has unit $\ell_2$-norm ($\|p_k\| = 1$) and fixed pairwise cosine similarity, where $\delta_{k,j}$ is the Kronecker delta:

$$p_k^\top p_j = \frac{K}{K-1} \delta_{k,j} - \frac{1}{K-1}, \quad \forall k, j \in [1, K], \tag{2}$$

**Neural Collapse Phenomenon.** The neural collapse phenomenon can be summarized as follows:

*(NC1) Feature Collapse*: Last-layer features of the same category collapse to their within-category mean: $\{\Sigma_W^{(k)} \to 0, \Sigma_W^{(k)} = \text{Avg}\{(\mu_{k,i} - \mu_k)(\mu_{k,i} - \mu_k)^\top\}$ where $\mu_{k,i}$ is the feature of the $i$-th sample in the category $k$ and $\mu_k$ is the within-category mean.

*(NC2) Convergence to Simplex ETF*: The within-category means of all categories, centered by the global mean $\mu_G = \text{Avg}_{k,i}(\mu_{k,i})$, converge to the vertices of a Simplex ETF: $\hat{\mu}_k = \mu_k - \mu_G / \|\mu_k - \mu_G\|, \forall k \in [1, K]$.

*(NC3) Self-Duality*: Within-category means align with the corresponding classifier weights: $\hat{\mu}_k = w_k / \|w_k\|$, $w_k$ is the weight of the category $k$.

*(NC4) Simplified Prediction*: Model prediction simplifies to a nearest-category-center rule: $\hat{y} = \arg\min_k \|\mu - \mu_k\| = \arg\max_k \langle \mu, w_k \rangle$, where $\mu$ is the feature of a test sample.

# 4 Method

## 4.1 Overview

As shown in Fig.2, our NC-GCD framework comprises four main components: a pre-trained visual encoder $f(\cdot)$, a periodic clustering module $g(\cdot)$, a pre-assigned ETF prototype set $P$, and a Semantic Consistency Matcher $\phi_{\text{SCM}}(\cdot)$.

First, we utilize the pre-trained visual encoder $f(\cdot)$ to extract the image embedding $e$ and augmented image embedding $e'$ for each sample. These embeddings are passed to the periodic clustering module $g(\cdot)$, which groups samples into clusters and computes the clustering prototypes $\{c_1, c_2, \ldots, c_K\}$. On the one hand, unsupervised ETF alignment pulls the top $\alpha\%$ of samples most similar to their cluster centers toward their corresponding pre-assigned ETF prototypes using a dot-regression loss.

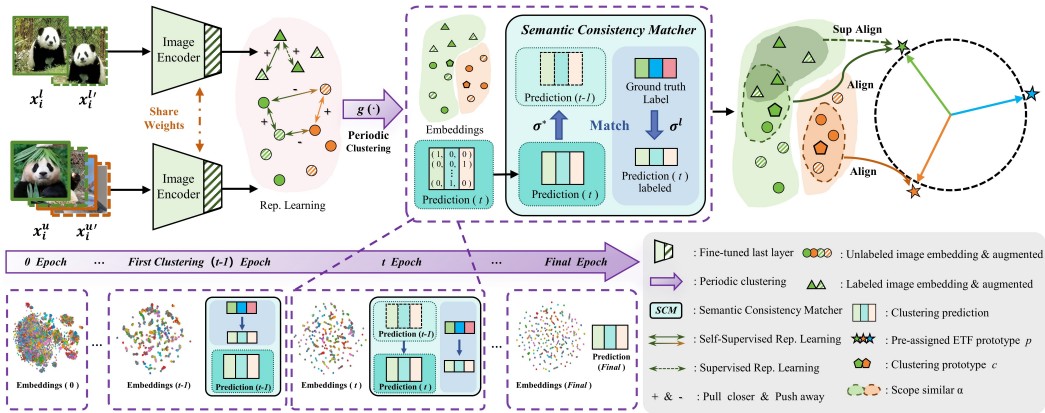

Figure 2: **Overview of the NC-GCD Framework**. The framework uses periodic clustering to group features, aligning them with pre-assigned ETF prototypes. The Semantic Consistency Matcher (SCM) ensures consistent label assignments across clustering iterations. This process stabilizes feature alignment and maintains a consistent geometric structure for both known and novel categories.

On the other hand, supervised ETF alignment ensures that labeled samples are aligned with their corresponding ETF prototypes, maintaining geometric consistency.

However, two key challenges arise. First, clustering is inherently unstable, leading to inconsistencies where the same category may be assigned to different clusters across iterations, making it difficult to maintain alignment with pre-assigned ETF prototypes. Second, in supervised alignment, the direct mapping between true labels and ETF prototypes may not always hold, creating mismatches that affect model stability. To address these issues, we introduce the Semantic Consistency Matcher (SCM), which guarantees one-to-one matching between clusters over time, reducing inconsistencies caused by clustering dynamics. By integrating these components, our method effectively mitigates category confusion and label imbalance, ensuring a structured feature space and improved performance.

### 4.2 ETF Alignment

**Pre-assigned ETF Prototype.** We construct a Simplex ETF as pre-assigned classifier prototypes to enforce a structured feature space and maximize category separability. Given or estimated the number of categories $K$, we can get its categories set $\mathcal{Y}_t$. Then we define the optimal ETF prototypes $\mathbf{P} = \{p_1, p_2, \ldots, p_K\}$ as Eq.1.

Each prototype $p_i \in \mathbb{R}^d$ maintains unit $\ell_2$-norm and follows a fixed pairwise similarity:

$$p_i^\top p_j = \frac{K}{K-1} q_i^\top q_j - \frac{1}{K-1}, \quad \forall i, j \in [1, K]. \tag{3}$$

Here, $q_i$ denotes a centered orthonormal basis of a $(K-1)$-dimensional space, with unit vectors satisfying $\sum_i q_i = \mathbf{0}$ and $q_i^\top q_j = \delta_{ij}$. By pre-assigning the ETF prototypes, we provide a stable and optimal alignment structure for both known and novel categories.

**Unsupervised ETF Alignment.** For each sample $x_i \in \mathcal{D}$, we first extract its image embedding $e_i = f(x_i)$ and augmented embedding $e_i'$. To introduce structural constraints and stabilize feature learning, we perform periodic clustering on the unlabeled dataset $\mathcal{D}$ every $T$ epochs. Given the extracted image embeddings $\{e_1, \ldots, e_N\}$, we apply clustering to obtain $K$ cluster centers $\{c_1, \ldots, c_K\}$. Then, each sample embedding $e_i$ is assigned to the nearest cluster based on the cosine similarity:

$$\hat{y}_i = \arg\max_k \frac{e_i^\top c_k}{\|e_k\|\|c_k\|}, \quad \forall x_i \in \mathcal{D}. \tag{4}$$

Once pseudo-labels $\hat{y}_i$ are assigned, we compute the similarity between each sample and its respective cluster center, where $\mathcal{D}_k$ is the set of samples assigned to cluster $k$:

$$s_i = \frac{e_i^\top c_k}{\|e_i\|\|c_k\|}, \quad \forall x_i \in \mathcal{D}_k. \tag{5}$$

To align features with the pre-assigned ETF prototype $\mathbf{P} = \{p_1, p_2, \ldots, p_K\}$, we select the top $\alpha\%$ of samples with the highest similarity $s_i$ within each cluster. These high-confidence samples are pulled toward the corresponding ETF prototype $p_k$ using a Dot-Regression Loss[30]:

$$\mathcal{L}_{\text{ETF}}^u = \frac{1}{|\tilde{D}_k|} \sum_{e_i \in \tilde{D}_k} \|e_i - p_k\|^2, \tag{6}$$

where $\tilde{D}_k = \{x_i \mid s_i \text{ is in the top } \alpha\% \text{ of } \mathcal{D}_k\}$ represents the selected high-confidence samples assigned to cluster $c_k$.

**Supervised ETF Alignment.** For each sample $x_i^l \in \mathcal{D}^l$, we extract its feature embedding $e_i^l$. The SCM establishes a one-to-one mapping between predicted clusters and ground-truth labels, transforming the original labels $y_i^l$ into ETF-aligned labels $y_i^{ETF} = \phi_{SCM}(y_i^l)$ (detailed in Section 4.3). Subsequently, each labeled sample is encouraged to align with its assigned ETF prototype:

$$\mathcal{L}_{\text{ETF}}^s = \frac{1}{|\mathcal{D}^l|} \sum_{x_i^l \in \mathcal{D}^l} \|e_i^l - p_a\|^2, a \in [1, K] \tag{7}$$

where $a = y_i^{ETF}$. This loss enforces geometric consistency between known category features and their corresponding ETF prototypes, ensuring a structured feature space.

**Consistent ETF Alignment Loss.** To balance the contributions of supervised and unsupervised ETF alignment, we define the hybrid ETF loss as:

$$\mathcal{L}_{\text{ETF}} = (1 - \gamma)\mathcal{L}_{\text{ETF}}^u + \gamma\mathcal{L}_{\text{ETF}}^s, \tag{8}$$

where $\gamma$ is a weighting coefficient determining.

### 4.3 Semantic Consistency Matcher (SCM)

In periodically clustering-based learning, two key challenges arise: 1) Pseudo-label instability across iterations. Clustering fluctuations introduce inconsistencies, disrupting the learning process and degrading feature alignment. 2) Misalignment between the predicted clusters and the true labels in supervised alignment. Supervised ETF alignment may misassign cluster labels, leading to further optimization instability. To mitigate these issues, we propose the SCM, which stabilizes pseudo-label assignments across iterations and ensures consistent alignment between predicted clusters and true labels. By enforcing a one-to-one mapping between clustering results from consecutive iterations, SCM reduces clustering-induced fluctuations.

Specifically, SCM aligns pseudo-labels between consecutive clustering iterations through an optimal assignment strategy. Given prediction sets $\{\hat{y}_1^t, \ldots, \hat{y}_N^t\}$, $\forall \hat{y}_i^t \in \mathcal{Y}_t$ for the current iteration and $\{\hat{y}_1^{t-1}, \ldots, \hat{y}_N^{t-1}\}$, $\forall \hat{y}_i^{t-1} \in \mathcal{Y}_{t-1}$ for the previous iteration, where $\mathcal{Y}_t$ and $\mathcal{Y}_{t-1}$ represent the corresponding label sets. Then, we seek an optimal permutation $\sigma^*$ that maximizes label consistency:

$$\sigma^* = \arg\max_{\sigma \in S_K} \sum_{k=1}^{K} \sum_{i=1}^{N} \mathbb{I}(\hat{y}_i^t = k)\mathbb{I}(\hat{y}_i^{t-1} = \sigma(k)), \tag{9}$$

where $\sigma$ is a bi-jective mapping function ensuring a one-to-one correspondence between clusters from consecutive iterations, and $S_K$ is the space of all valid permutations over $K$ clusters. The indicator function $\mathbb{I}(\cdot)$ returns 1 if the condition holds and 0 otherwise. The label set in the current iteration is updated as $\mathcal{Y}_t = \sigma^*(\mathcal{Y}_{t-1})$.

SCM also ensures consistency in supervised ETF alignment by optimally mapping predicted clusters to their corresponding ETF-aligned labels. Given the prediction of the labeled samples and the set of ground truth labels, SCM applies the same optimal assignment strategy to obtain the permutation $\sigma^l$. Using this mapping, the ETF-aligned label set is updated as: $\mathcal{Y}_t^{ETF} = \sigma^l(\mathcal{Y}^l)$.

### 4.4 Comparative Representation Learning

Following CMS[8], we adopt the unsupervised loss encourages similarity between embeddings of the same sample while distinguishing them from other samples:

$$\mathcal{L}_{\text{REP}}^u = -\frac{1}{|B|} \sum_{i \in B} \log \frac{\exp(e_i \cdot e_i'/\tau)}{\sum_{j \neq i} \exp(e_i \cdot e_j/\tau)}, \tag{10}$$

Table 1: Comparison with the state of the arts on GCD, evaluated with or without the GT $K$ for clustering, with DINOv1 backbone.

| Method | CUB | | | Stanford Cars | | | FGVC Aircraft | | | Herbarium 19 | | | CIFAR100 | | | ImageNet100 | | |
|---|---|---|---|---|---|---|---|---|---|---|---|---|---|---|---|---|---|---|
| | All | Old | New | All | Old | New | All | Old | New | All | Old | New | All | Old | New | All | Old | New |
| *(a) Clustering with the ground-truth number of categories $K$ given* | | | | | | | | | | | | | | | | | | |
| GCD (CVPR22) | 51.3 | 56.6 | 48.7 | 39.0 | 57.6 | 29.9 | 45.0 | 41.1 | 46.9 | 35.4 | 51.0 | 27.0 | 73.0 | 76.2 | 66.5 | 74.1 | 89.8 | 66.3 |
| DCCL (CVPR23) | 63.5 | 60.8 | 64.9 | 43.1 | 55.7 | 36.2 | - | - | - | - | - | - | 75.3 | 76.8 | 70.2 | 80.5 | 90.5 | 76.2 |
| PromptCAL | 62.9 | 64.4 | 62.1 | 50.2 | 70.1 | 40.6 | 52.2 | 52.2 | 52.3 | 37.0 | 52.0 | 28.9 | 81.2 | 84.2 | 75.3 | 83.1 | 92.7 | 78.3 |
| GPC (ICCV23) | 55.4 | 58.2 | 53.1 | 42.8 | 59.2 | 32.8 | 46.3 | 42.5 | 47.9 | - | - | - | 77.9 | 85.0 | 63.0 | 76.9 | 94.3 | 71.0 |
| SimGCD (ICCV23) | 60.3 | 65.6 | 57.7 | 53.8 | 71.9 | 45.0 | 54.2 | 59.1 | 51.8 | 44.0 | 58.0 | 36.4 | 80.1 | 81.2 | **77.8** | 83.0 | 93.1 | 77.9 |
| PIM (CVPR23) | 62.7 | 75.7 | 56.2 | 43.1 | 66.9 | 31.6 | - | - | - | 42.3 | 56.1 | 34.8 | 78.3 | 84.2 | 66.5 | 83.1 | 95.3 | 77.0 |
| TRAILER (CVPR24) | 65.1 | 71.3 | 61.9 | 55.4 | 71.7 | 47.6 | 54.5 | 62.6 | 50.5 | 44.5 | 57.0 | 37.8 | - | - | - | - | - | - |
| SelEx (ECCV24) | 73.6 | 75.3 | 72.8 | 58.5 | 75.6 | 50.3 | 57.1 | **64.7** | 53.3 | - | - | - | 82.3 | 85.3 | 76.3 | 83.1 | 93.6 | 77.8 |
| CMS (CVPR24) | 68.2 | 76.5 | 64.0 | 56.9 | 76.1 | 47.6 | 56.0 | 63.4 | 52.3 | 36.4 | 54.9 | 26.4 | 82.3 | **85.7** | 75.5 | 84.7 | **95.6** | 79.2 |
| SPT (ICLR24) | 65.8 | 68.8 | 65.1 | 59.0 | **79.2** | 49.3 | 59.3 | 61.8 | 58.1 | 43.4 | **58.7** | 35.2 | 81.3 | 84.3 | 75.6 | 85.4 | 93.2 | 81.4 |
| **Ours (NC-GCD)** | **74.8** | 76.8 | **73.8** | **59.9** | 77.8 | **51.2** | **60.0** | 57.6 | **61.2** | **46.4** | 58.4 | **40.7** | **82.7** | 85.5 | 77.3 | **88.4** | 94.1 | **85.5** |
| *(b) Clustering without the ground-truth number of categories $K$ given* | | | | | | | | | | | | | | | | | | |
| GCD (CVPR22) | 51.1 | 56.4 | 48.4 | 39.1 | 58.6 | 29.7 | - | - | - | 37.2 | 51.7 | 29.4 | 70.8 | 77.6 | 57.0 | 77.9 | 91.1 | 71.3 |
| GPC (ICCV23) | 52.0 | 55.5 | 47.5 | 38.2 | 58.9 | 27.4 | 43.3 | 40.7 | 44.8 | 36.5 | 51.7 | 27.9 | 75.4 | **84.6** | 60.1 | 75.3 | 93.4 | 66.7 |
| PIM (CVPR23) | 62.0 | **75.7** | 55.1 | 42.4 | 65.3 | 31.3 | - | - | - | 42.0 | 55.5 | 34.7 | 75.6 | 81.6 | 63.6 | 83.0 | 95.3 | 76.9 |
| CMS (CVPR24) | 64.4 | 68.2 | 62.2 | 51.7 | 68.9 | 43.4 | 55.2 | **60.6** | 52.4 | 37.4 | **56.5** | 27.1 | 79.6 | 83.2 | 72.3 | 81.3 | 95.6 | 74.2 |
| **Ours (NC-GCD)** | **70.3** | 72.1 | **69.4** | **54.0** | **73.1** | **44.8** | **55.4** | 57.3 | **54.5** | **42.3** | 56.2 | **34.8** | **80.5** | 83.7 | **74.0** | **85.7** | **95.9** | **80.6** |

where $e_i'$ is the augmented view of $e_i$, $\tau$ is the temperature parameter, and $|B|$ is the batch size.

For labeled samples, we apply a supervised loss to enforce compact intra-category representations:

$$\mathcal{L}_{\text{REP}}^{s} = -\frac{1}{|B_l|} \sum_{i \in B_l} \frac{1}{|H(i)|} \sum_{h \in H(i)} \log \frac{\exp(e_i^l \cdot e_h / \tau)}{\sum_{j \neq i} \exp(e_i^l \cdot e_j / \tau)}, \tag{11}$$

where $H(i)$ represents the samples sharing the same label $i$. The representation learning objective is:

$$\mathcal{L}_{\text{REP}} = (1 - \lambda)\mathcal{L}_{\text{REP}}^{u} + \lambda\mathcal{L}_{\text{REP}}^{s}, \tag{12}$$

where $\lambda$ balances the contributions of unsupervised and supervised components.

### 4.5 Final Objective

The final loss function integrates all components to ensure stable feature alignment and category separability:

$$\mathcal{L} = \beta\mathcal{L}_{\text{ETF}} + \mathcal{L}_{\text{REP}}, \tag{13}$$

where $\beta$ controls the contributions of ETF loss and representation learning loss.

## 5 Experiments

### 5.1 Experimental Setup

**Datasets.** Our evaluation spans six image classification benchmarks, including both generic and fine-grained datasets. For generic datasets, we use CIFAR-100 [37] and ImageNet-100 [38]. For fine-grained datasets, we evaluate on CUB-200 [39], Stanford Cars [40], FGVC Aircraft [41], and Herbarium19 [42]. To separate categories into known and novel categories, we follow the SSB split protocol [43] for the fine-grained datasets. For CIFAR-100 and ImageNet-100, we perform a random category split using a fixed seed, consistent with previous studies.

**Implementation Details.** Our method utilizes the pre-trained DINO ViT-B/16 [44, 45] as the backbone network. We fine-tune the final layer of the image encoder along with a projection head, following prior work [4, 8]. The projection head consists of an MLP with a 768-dimensional input, a 2048-dimensional hidden layer, and a 768-dimensional output, followed by a GeLU activation function [46]. The learning rate is set to 0.1. Other hyperparameters, including batch size, temperature

Table 2: Comparison of various methods across multiple metrics, including Fine-grained datasets, Classification datasets, and All datasets.

| Method | Fine-grained Avg | | | Classification Avg | | | All Avg | | |
|---|---|---|---|---|---|---|---|---|---|
| | All | Old | New | All | Old | New | All | Old | New |
| *(a) Clustering with the ground-truth number of categories K given* | | | | | | | | | |
| SimGCD (ICCV 2023) | 53.1 | 63.7 | 47.7 | 81.6 | 87.2 | 77.9 | 62.6 | 71.5 | 57.8 |
| CMS (CVPR 2024) | 54.4 | **67.7** | 47.6 | 83.5 | **90.7** | 77.4 | 64.1 | **75.4** | 57.5 |
| SPT (ICLR 2024) | 56.9 | 67.1 | 51.9 | 83.4 | 88.8 | 78.5 | 65.7 | 74.3 | 60.8 |
| **Ours (NC-GCD)** | **60.3** | 67.6 | 56.7 | **85.5** | 89.8 | **81.4** | **68.7** | 75.0 | **64.9** |
| *Improv. over SPT* | **+3.4** | **+0.5** | **+4.8** | **+2.1** | **+1.0** | **+2.9** | **+3.0** | **+0.7** | **+4.1** |
| *(b) Clustering without the ground-truth number of categories K given* | | | | | | | | | |
| GPC (ICCV 2023) | 46.2 | 50.8 | 38.9 | 75.4 | 89.0 | 63.4 | 53.5 | 64.1 | 45.7 |
| CMS (CVPR 2024) | 52.2 | 63.6 | 46.3 | 80.5 | 89.4 | 73.3 | 61.6 | 72.2 | 55.3 |
| **Ours (NC-GCD)** | **55.5** | **64.6** | **50.9** | **83.1** | **89.8** | **77.3** | **64.7** | **73.0** | **59.7** |
| *Improv. over CMS* | **+3.3** | **+1.0** | **+4.6** | **+2.6** | **+0.4** | **+4.0** | **+3.1** | **+0.8** | **+4.4** |

$\tau_s$, weight decay $\lambda$, and the number of augmentations, are set to 128, 0.07, $1e^{-4}$, and 2, respectively, in accordance with previous studies. All experiments are conducted on an NVIDIA 3090 GPU. Further implementation details can be found in the Appendix.

## 5.2 Comparison with State of the Art Methods

We evaluate our NC-GCD framework on several fine-grained and generic classification benchmarks to demonstrate its efficacy in GCD. Table 1 presents a comparison between our method and the previous methods [4, 5, 6, 12, 13, 7, 8, 47, 10, 36], evaluated on both fine-grained and generic datasets with and without access to the ground-truth (GT) number of categories $K$ for clustering. For the case without access to GT $K$, we estimate the $K$ through clustering, which is then used to align the ETF prototypes. Meanwhile, to comprehensively evaluate the performance of various methods across Fine-Grained Datasets, Generic Classification Datasets, and All datasets, Table 2 presents the average performance.

**Fine-Grained Datasets.** On fine-grained benchmarks such as CUB-200, Stanford Cars, FGVC Aircraft, and Herbarium19, NC-GCD outperforms previous methods, whether or not the ground-truth $K$ is provided. As shown in Table 2a, when the GT $K$ is available, NC-GCD achieves an average all-category accuracy of 60.3%, surpassing SPT by **3.4%** and improving novel category accuracy over SPT by **4.8%**. Even without GT $K$, as shown in Table 2b, NC-GCD remains robust, surpassing CMS by **3.3%** in all-category accuracy. Notably, on CUB-200, NC-GCD outperforms CMS by **5.9%** in all-category accuracy and **7.2%** in novel category accuracy, demonstrating strong generalization without the number of categories. Fine-grained datasets present higher category confusion due to minimal intra-category variation, and our method effectively mitigates this and enhances category separability.

**Generic Classification Datasets.** NC-GCD achieves strong performance on standard classification datasets such as CIFAR-100 and ImageNet-100, as shown in Table 2. With known $K$, NC-GCD attains an all-category accuracy of **82.7%** on CIFAR-100 and **88.4%** on ImageNet-100, outperforming CMS and SPT. Notably, novel category accuracy improves by **1.7%** on CIFAR-100 and **4.1%** on ImageNet-100. When $K$ must be estimated, NC-GCD maintains its competitive edge, achieving **85.7%** all-category accuracy on ImageNet-100, surpassing CMS by **4.4%**, demonstrating its adaptability in generic classification scenarios.

Across both fine-grained and generic datasets, NC-GCD consistently achieves well performance. As shown in Table 2a, with GT $K$, NC-GCD reaches the highest all-category accuracy of **68.7%**, surpassing SPT by **3.0%** and improving novel category accuracy by **4.1%**. Even without GT $K$, NC-GCD remains robust, surpassing CMS by **3.1%** in all-category accuracy and **4.4%** in novel category accuracy. By pre-assigning ETF prototypes, NC-GCD effectively improves category separability, mitigating category confusion and ensuring balanced learning across old and novel categories. These results highlight the robustness and versatility of NC-GCD for real-world GCD applications.

Table 3: Performance comparison with Supervised and Unsupervised ETF Alignment.

| Sup ETF Alignment | Unsup ETF Alignment | CUB | | | Herbarium 19 | | | ImageNet100 | | | Six Datasets Avg | | |
|---|---|---|---|---|---|---|---|---|---|---|---|---|---|
| | | All | Old | New | All | Old | New | All | Old | New | All | Old | New |
| ✗ | ✗ | 67.7 | 75.7 | 63.9 | 36.5 | 55.0 | 26.5 | 84.4 | 94.0 | 80.8 | 63.3 | 73.7 | 57.5 |
| ✓ | ✗ | 69.6 | 75.8 | 66.5 | 37.6 | 56.7 | 28.1 | 85.0 | **95.3** | 79.3 | 64.6 | **75.2** | 58.3 |
| ✗ | ✓ | **75.7** | 71.3 | **77.8** | **47.2** | **60.0** | 40.3 | 87.6 | 94.5 | 84.1 | 68.0 | 73.2 | 64.9 |
| ✓ | ✓ | 74.8 | 76.8 | 73.8 | 46.4 | 58.4 | **40.7** | **88.4** | 94.1 | **85.5** | **68.7** | 74.9 | 64.9 |
| *Improv. over baseline* | | **+7.1** | **+1.1** | **+9.9** | **+9.9** | **+3.4** | **+14.2** | **+4.0** | **+0.1** | **+4.7** | **+5.4** | **+1.2** | **+7.4** |

## 5.3 Ablation Study

**The Effectiveness of ETF Alignment.** To evaluate the impact of *Unsupervised ETF Alignment* and *Supervised ETF Alignment*, we conduct ablation studies with four configurations: (1) Baseline (both disabled), (2) Supervised ETF only, (3) Unsupervised ETF only, and (4) Full model (both enabled). Table 3 presents the results, revealing key insights into each component's contribution.

*Unsupervised ETF Alignment:* This component yields the most significant improvements, especially in fine-grained datasets. As shown in Table 3, enabling unsupervised ETF alignment alone leads to substantial gains in novel category accuracy. The model effectively learns robust representations, particularly for novel categories, where differentiation between visually similar categories is crucial.

*Supervised ETF Alignment:* This component stabilizes and preserves old categories' accuracy. By aligning labeled features with their corresponding ETF prototypes, it ensures the retention of learned knowledge while maintaining high performance on known categories. Though its contribution to novel category accuracy is smaller than that of unsupervised alignment, it plays a crucial role in preventing forgetting and maintaining model stability.

*Synergistic Combination:* The full model, integrating both alignments, achieves the highest overall accuracy, with an average all-category accuracy of **68.7%**, significantly surpassing the baseline. This combination enhances novel category discovery while preserving old category performance. Specifically, the average novel category accuracy increases by **7.4%**, and the overall accuracy improves by **5.4%**. The results highlight the synergy between supervised and unsupervised alignment, demonstrating the NC-GCD effectiveness in balancing learning across known and novel categories.

**The Effectiveness of SCM.** To evaluate the impact of the Semantic Consistency Matcher (SCM), we conduct ablation studies on CUB-200, ImageNet100, and Herbarium 19. As shown in Table 4, enabling SCM consistently improves performance on all three datasets. On Herbarium 19, which poses greater challenges due to its large number of categories and high intra-category variance, SCM provides more

Table 4: Ablation study of our SCM module.

| SCM | CUB | | | Herbarium 19 | | | ImageNet100 | | |
|---|---|---|---|---|---|---|---|---|---|
| | All | Old | New | All | Old | New | All | Old | New |
| ✗ | 70.3 | 71.0 | 70.0 | 42.6 | 54.3 | 36.3 | 84.3 | 88.9 | 82.0 |
| ✓ | **75.7** | **71.3** | **77.8** | **47.2** | **60.0** | **40.3** | **87.6** | **94.5** | **84.1** |
| *Improv.* | **+5.4** | **+0.3** | **+7.8** | **+4.6** | **+5.7** | **+4.0** | **+3.3** | **+5.6** | **+2.1** |

significant improvements, with gains of **4.6%** for all categories and **5.7%** for novel categories, demonstrating its effectiveness in complex classification tasks. By mitigating clustering fluctuations, SCM ensures stable pseudo-label assignments. These results highlight its crucial role in stabilizing training, particularly in datasets with large category numbers.

**The Influence of the ETF Scope Coefficient $\alpha$.** The ETF Scope Coefficient $\alpha$ plays a crucial role in determining the number of samples selected for alignment with the ETF prototype. By controlling the percentage of high-confidence samples, $\alpha$ influences the model's ability to focus on either old or novel categories. As seen in Fig. 3, when $\alpha$ is too small, the model may not leverage enough high-confidence samples, leading to suboptimal performance. On the other hand, as $\alpha$ increases, novel category accuracy improves significantly,

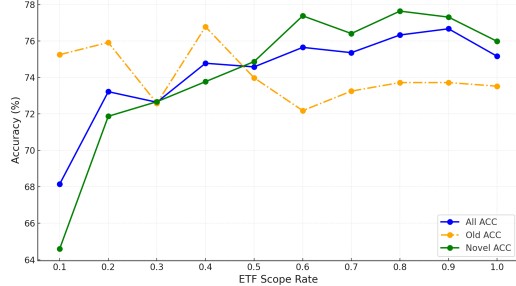

Figure 3: Ablation study of the $\alpha$ on CUB-200.

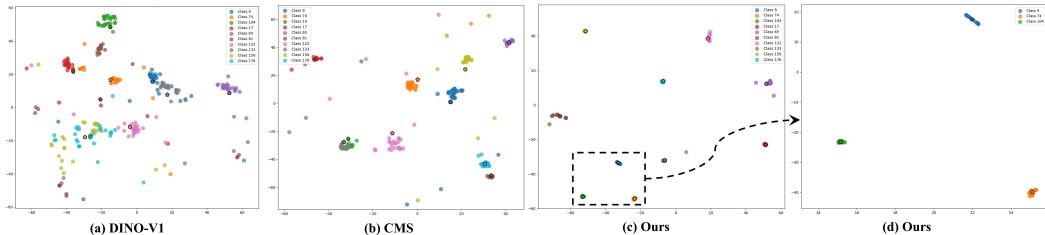

Figure 4: Visualization of Comparison with DINOv1, CMS, and our method on CUB-200.

especially in fine-grained datasets where category distinctions are subtle. However, when $\alpha$ exceeds a certain threshold, the model's focus on new categories comes at the expense of old category accuracy, highlighting the trade-off between the two. The best balance is achieved at $\alpha = 0.8$, where both old and novel category accuracy reach optimal values, demonstrating the importance of selecting an appropriate $\alpha$.

**Ablation study of *ETF loss coefficient* $\beta$ , more results of the effectiveness of ETF Alignment and *Parameter analysis* will be presented in the Appendix.**

### 5.4 Visualization

**Comparison with Previous Methods.** As shown in Fig. 4, we compare the embeddings extracted by DINOv1 (a), CMS (b), and our method (c) for ten categories, with (d) focusing on a subset of three categories from (c). The embeddings from DINO-V1 and CMS show significant overlap between categories, whereas our method clearly separates the categories into distinct clusters. Notably, categories 17, 74, and 104 collapse almost into a single point in (c), which aligns with the Neural Collapse hypothesis. This demonstrates that our method enhances feature discriminability, improving category separability across both known and novel categories.

## 6 Conclusion and Future Work

We propose NC-GCD, a framework for Generalized Category Discovery that integrates unsupervised and supervised ETF alignment to enhance category separability. Extensive experiments demonstrate that NC-GCD consistently outperforms previous methods in both known and novel category accuracy. Ablation studies validate the complementary effects of the two alignment strategies, while t-SNE visualizations highlight its effectiveness in distinguishing similar categories, aligning with the Neural Collapse hypothesis. A promising future research direction is extending our NC-GCD to incremental GCD settings.

## 7 Acknowledgements

This work is supported by the National Natural Science Foundation of China under Grant No.U21B2048 and No.62302382, the Shenzhen Key Technical Projects Under Grant CJGJZD2022051714160501, the China Postdoctoral Science Foundation No.2024M752584, and the Joint Fund Project of the National Natural Science Foundation of China under Grant No.U24A20328.

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

# Appendix Contents

# A    More Implementation Details

## A.1    Training Parameters and Details

In this study, we evaluate the NC-GCD framework on various datasets using a pre-trained DINO ViT-B/16 model as the backbone. The final layer of the image encoder is fine-tuned alongside a projection head for both supervised and unsupervised alignment tasks, as per previous works [4, 8]. The projection head is a Multi-Layer Perceptron (MLP) consisting of a 768-dimensional input layer, a 2048-dimensional hidden layer, and a 768-dimensional output layer, followed by a GeLU activation function [46]. We empirically set periodic clustering epoch T=5 as a trade-off between stability and computational efficiency. Larger T may reduce adaptation speed, while smaller T increases computational cost.

The learning rate is set to 0.1, and other key hyperparameters such as batch size (128), temperature ($\tau_s = 0.07$), weight decay ($1e^{-4}$), and the number of augmentations (2) follow the settings used in prior works[4]. All experiments are conducted using an NVIDIA 3090 GPU, which provides sufficient power for the fine-grained operations required in our method.

For training, we use the SGD optimizer with a momentum of 0.9 and weight decay of $1e^{-4}$. We apply a cosine annealing learning rate scheduler to adapt the learning rate during the course of training. The method is trained for a total of 200 epochs. Additionally, data augmentation and random cropping are applied to enhance the model's ability to generalize across different transformations.

In our model, we combine supervised and unsupervised contrastive losses and the ETF alignment loss. The unsupervised ETF alignment uses a clustering method for periodic feature grouping, while the supervised ETF alignment ensures that labeled data is aligned with their corresponding ETF prototypes. During training, we use a weighted random sampler to balance labeled and unlabeled data in each batch, allowing the model to learn from both known and novel categories effectively.

## A.2    Dataset Details and Category Splits

We evaluate our proposed framework on six diverse datasets covering general and fine-grained image classification tasks. Table 5 provides an overview of the category distributions for each dataset, categorizing them as known and unknown categories for assessing Generalized Category Discovery (GCD) performance. Below, we summarize the details and category split protocols for each dataset.

**CIFAR-100.**    CIFAR-100 [37] consists of 60,000 images across 100 categories. For the GCD evaluation, we used 80 categories as known categories for training and the remaining 20 as unknown categories for testing.

**ImageNet100.** ImageNet100 [38] is a subset of ImageNet with 100 categories. Following standard protocols, we randomly assign 50 categories as known categories and the other 50 as unknown categories.

**CUB-200-2011.** CUB-200-2011 [39] is a fine-grained bird dataset with 200 categories. We split the dataset into 100 known and 100 unknown categories following the standard fine-grained GCD protocol [43].

**Stanford Cars.** Stanford Cars [40] is a fine-grained dataset with 196 car categories. We use 98 categories for training (known categories) and 98 for testing (unknown categories), consistent with previous fine-grained GCD work.

Table 5: Distribution of labeled (known) and unlabeled (novel) categories across datasets. Labeled categories have annotated training samples and serve as known categories for supervision, while unlabeled categories represent novel categories that the model must discover without labels.

| Dataset | Labeled Categories | Unlabeled Categories |
|---|---|---|
| CIFAR100 [37] | 80 | 20 |
| ImageNet100 [38] | 50 | 50 |
| CUB-200-2011 [39] | 100 | 100 |
| Stanford Cars [40] | 98 | 98 |
| FGVC-Aircraft [41] | 50 | 50 |
| Herbarium19 [42] | 341 | 342 |

**FGVC Aircraft.** FGVC Aircraft [41] includes 100 aircraft categories. The dataset is split into 50 known and 50 unknown categories, adhering to the SSB protocol.

---

**Algorithm 1** Semantic Consistency Matcher (SCM)

---

1: **Input:**
   - Prediction sets from the current iteration: $\{\hat{y}_1^t, \ldots, \hat{y}_N^t\}, \quad \forall \hat{y}_i^t \in \mathcal{Y}_t$
   - Prediction sets from the previous iteration: $\{\hat{y}_1^{t-1}, \ldots, \hat{y}_N^{t-1}\}, \quad \forall \hat{y}_i^{t-1} \in \mathcal{Y}_{t-1}$
   - Prediction sets of labeled samples from the current iteration: $\{\hat{y}_1^{tl}, \ldots, \hat{y}_M^{tl}\}, \quad \forall \hat{y}_i^t \in \mathcal{Y}_t^l$
   - Ground-truth labels sets: $\{y_1^l, \ldots, y_M^l\}, \quad \forall y_i^l \in \mathcal{Y}^l$
   - Number of clusters: $K$
2: **Output:**
   - Updated label set: $\mathcal{Y}_t \leftarrow \sigma^*(\mathcal{Y}_{t-1})$
   - Updated supervised ETF-aligned label set: $\mathcal{Y}_t^{ETF} \leftarrow \sigma^l(\mathcal{Y}^l)$
3: **Define permutation set:**

$$S_K = \{\sigma \mid \sigma : \{1, 2, \ldots, K\} \to \{1, 2, \ldots, K\}, \sigma \text{ is bi-jective mapping function}\}$$

4: **Step 1: Optimal Alignment for Prediction (Unsupervised)**
5:       Find an optimal one-to-one mapping $\sigma^*$ between clusters at iteration $t-1$ and $t$:

$$\sigma^* = \arg \max_{\sigma \in S_K} \sum_{k=1}^{K} \sum_{i=1}^{N} \mathbb{I}(\hat{y}_i^t = k)\mathbb{I}(\hat{y}_i^{t-1} = \sigma(k))$$

6:       Here, $S_K$ denotes the set of all possible permutations (bi-jective mappings) over $K$ clusters
7:       Update pseudo-labels for the current iteration:

$$\mathcal{Y}_t \leftarrow \sigma^*(\mathcal{Y}_{t-1})$$

8: **Step 2: Optimal Alignment for Supervised ETF Labels**
9:       Find an optimal one-to-one mapping $\sigma^l$ between predicted labels and ground-truth labels:

$$\sigma^l = \arg \max_{\sigma \in S_J} \sum_{j=1}^{J} \sum_{i=1}^{M} \mathbb{I}(\hat{y}_i^{tl} = j)\mathbb{I}(y_i^l = \sigma(j))$$

10:      Here, $S_J$ denotes the set of all possible permutations (bi-jective mappings) over $J$ clusters
11:      Update ETF-aligned labels for supervised samples:

$$\mathcal{Y}_t^{ETF} \leftarrow \sigma^l(\mathcal{Y}^l)$$

12: **Return:** Updated label set $\mathcal{Y}_t$ and ETF-aligned supervised label set $\mathcal{Y}_t^{ETF}$

---

**Herbarium19.** Herbarium19 [42] is a large-scale fine-grained dataset with 683 plant species. Due to its high intra-category variance, we split the dataset into 341 known and 342 unknown categories to evaluate GCD on more challenging fine-grained datasets.

**Category Split Protocol.** For general datasets (CIFAR-100 and ImageNet100), we apply a random category split using a fixed seed to ensure reproducibility. For fine-grained datasets (CUB-200-2011, Stanford Cars, and FGVC Aircraft), we use the SSB split protocol [43], which ensures balanced category distributions while preserving intra-category variance. For Herbarium19, the dataset is split into nearly equal known and unknown categories, testing the model's robustness in handling datasets with significant intra-category variation.

## A.3 Pseudocode of SCM

Semantic Consistency Matcher (SCM) addresses two key issues in generalized category discovery tasks: the instability of pseudo-label assignments caused by periodic clustering and the misalignment between predicted labels and ground-truth labels in supervised ETF alignment. Specifically, SCM operates in two steps, as detailed in Algorithm 1.

**(1) Unsupervised Scenario (Pseudo-label Alignment):** Given a prediction set from the current iteration with $N$ samples, let $\{\hat{y}_1^t, \ldots, \hat{y}_N^t\}$, where $\hat{y}_i^t \in \mathcal{Y}_t$. Similarly, prediction sets from the

Table 6: Performance comparison on CUB, Stanford Cars, FGVC Aircraft, and Herbarium 19.

| Sup ETF Alignment | Unsup ETF Alignment | CUB | | | Stanford Cars | | | FGVC Aircraft | | | Herbarium 19 | | |
|---|---|---|---|---|---|---|---|---|---|---|---|---|---|
| | | All | Old | New | All | Old | New | All | Old | New | All | Old | New |
| ✗ | ✗ | 67.7 | 75.7 | 63.9 | 55.1 | 74.3 | 45.8 | 54.3 | 57.4 | 52.8 | 36.5 | 55.0 | 26.5 |
| ✓ | ✗ | 69.6 | 75.8 | 66.5 | 57.0 | 75.1 | 46.7 | 56.1 | 62.9 | 53.2 | 37.6 | 56.7 | 28.1 |
| ✗ | ✓ | 75.7 | 71.3 | 77.8 | 56.0 | 74.7 | 46.9 | 59.8 | 54.7 | 62.3 | 47.2 | 60.0 | 40.3 |
| ✓ | ✓ | 74.8 | 76.8 | 73.8 | 59.9 | 77.8 | 51.2 | 60.0 | 57.6 | 61.2 | 46.4 | 58.4 | 40.7 |
| *Improv. over baseline* | | **+7.1** | **+1.1** | **+9.9** | **+4.8** | **+3.5** | **+5.4** | **+5.7** | **+0.2** | **+8.4** | **+9.9** | **+3.4** | **+14.2** |

Table 7: Performance comparison on CIFAR100, ImageNet100, and Six Datasets Average results.

| Sup ETF Alignment | Unsup ETF Alignment | CIFAR100 | | | ImageNet100 | | | Six Datasets Avg | | |
|---|---|---|---|---|---|---|---|---|---|---|
| | | All | Old | New | All | Old | New | All | Old | New |
| ✗ | ✗ | 82.0 | 85.4 | 75.2 | 84.4 | 94.0 | 80.8 | 63.3 | 73.7 | 57.5 |
| ✓ | ✗ | 82.3 | **85.7** | 75.7 | 85.0 | **95.3** | 79.3 | 64.6 | **75.2** | 58.3 |
| ✗ | ✓ | 82.0 | 84.1 | **77.8** | 87.6 | 94.5 | 84.1 | 68.0 | 73.2 | 64.9 |
| ✓ | ✓ | **82.7** | 85.5 | 77.3 | **88.4** | 94.1 | **85.5** | **68.7** | 74.9 | 64.9 |
| *Improv. over baseline* | | **+0.7** | **+0.1** | **+2.1** | **+4.0** | **+0.1** | **+4.7** | **+5.4** | **+1.2** | **+7.4** |

previous iteration denote as: $\{\hat{y}_1^{t-1}, \ldots, \hat{y}_N^{t-1}\}$, where $\hat{y}_i^{t-1} \in \mathcal{Y}_{t-1}$. Both $\mathcal{Y}_t$ and $\mathcal{Y}_{t-1}$ contain $K$ cluster labels. SCM finds an optimal bijective mapping $\sigma^*$ between clusters of consecutive iterations by maximizing their agreement. $S_K$ denotes all possible permutations (one-to-one mappings) over $K$ clusters. The indicator function $\mathbb{I}(\cdot)$ returns 1 if the condition is satisfied and 0 otherwise. The pseudo-label set at iteration $t$ is then updated as: $\mathcal{Y}_t = \sigma^*(\mathcal{Y}_{t-1})$.

**(2) Supervised Scenario:** In the supervised setting, given $M$ labeled samples, $\{\hat{y}_1^{tl}, \ldots, \hat{y}_M^{tl}\}$ denote the predicted sets of labels samples from the current iteration $t$. Let $\{y_1^l, \ldots, y_M^l\}$ denote the ground-truth labels, each belonging to the true label set $\mathcal{Y}^l$ with cardinality $J$. SCM identifies an optimal one-to-one mapping $\sigma^l$ between predicted and ground-truth labels by maximizing their consistency. $S_J$ denotes all possible permutations over the $J$ known categories. The supervised ETF-aligned label set is then updated as: $\mathcal{Y}_t^{\text{ETF}} = \sigma^l(\mathcal{Y}^l)$.

# B More Experiment Results and Ablation Study

## B.1 Detailed Experiment Results of the Effectiveness of ETF Alignment

In the main manuscript, we have demonstrated the effectiveness of the proposed ETF Alignment strategy on CUB, Herbarium 19, and ImageNet100 datasets. To further validate its generalization capability, we conduct additional experiments on Stanford Cars, FGVC Aircraft, and CIFAR100, and report the comprehensive results in Tables 6 and 7.

From Table 6, we observe that applying either supervised or unsupervised ETF Alignment leads to noticeable performance improvements across different datasets. Specifically, the combination of both supervised and unsupervised ETF Alignment consistently achieves the best results, especially on the novel categories, which demonstrates the effectiveness of our method in handling new categories under a generalized category discovery scenario.

Table 7 further reports the results on CIFAR100 and ImageNet100, as well as the average performance over all six datasets. Our method achieves significant improvements in both overall accuracy and novel category accuracy, confirming its strong generalization ability and robustness across various datasets and domains.

## B.2 Main Results with Different Backbones

To evaluate the generalization ability of NC-GCD across different pretrained representations, we conduct experiments using three representative backbones: DINOv1, DINOv2, and CLIP. The results on the CUB-200 dataset are summarized in Table 8.

Table 8: Comparison across different backbones (DINOv1, DINOv2, CLIP) and average on CUB200. The best results are in bold, the second best are underlined.

| Method | DINOv1 | | | DINOv2 | | | CLIP | | | AVG | | |
|---|---|---|---|---|---|---|---|---|---|---|---|---|
| | ALL | Old | New | ALL | Old | New | ALL | Old | New | ALL | Old | New |
| GCD (CVPR 22) | 51.3 | 56.6 | 48.7 | 71.9 | 71.2 | 72.3 | 51.1 | 56.2 | 48.6 | 58.1 | 61.3 | 56.5 |
| SimGCD (ICCV 23) | 60.3 | 65.6 | 57.7 | 74.9 | 78.5 | 73.1 | 69.6 | 75.8 | 66.5 | 68.3 | 73.3 | 65.8 |
| ProtoGCD (TPAMI-25) | 63.5 | 68.5 | 60.5 | 75.7 | 81.5 | 72.9 | - | - | - | - | - | - |
| RLCD (ICML 25) | 70.0 | **79.1** | 65.4 | 78.7 | 79.5 | 78.3 | - | - | - | - | - | - |
| SelEx (ECCV 24) | 73.6 | 75.3 | 72.8 | 87.4 | 85.1 | 88.5 | 74.2 | 69.5 | 76.5 | 78.4 | 76.6 | 79.3 |
| **Ours (NC-GCD)** | **74.8** | 76.8 | **73.8** | **87.9** | **85.3** | **89.2** | **78.5** | **79.3** | **78.0** | **80.4** | **80.5** | **80.3** |
| *Improvement over SelEx* | *+1.2* | *+1.5* | *+1.0* | *+0.5* | *+0.2* | *+0.7* | *+4.3* | *+9.8* | *+1.5* | *+2.0* | *+3.9* | *+1.0* |

Table 9: Performance on CUB-200 with different ETF loss rates.

| ETF loss rate $\beta$ | CUB-200 | | | | | |
|---|---|---|---|---|---|---|
| | Best ACC | | | Final ACC | | |
| | ALL | Old | Novel | ALL | Old | Novel |
| 0.5 | 71.57 | **75.85** | 69.44 | 71.44 | **75.85** | 69.24 |
| 1 | 76.67 | 73.72 | 77.31 | **76.33** | 74.18 | **77.41** |
| 2 | **77.45** | 74.85 | **78.75** | 71.42 | 66.11 | 74.07 |

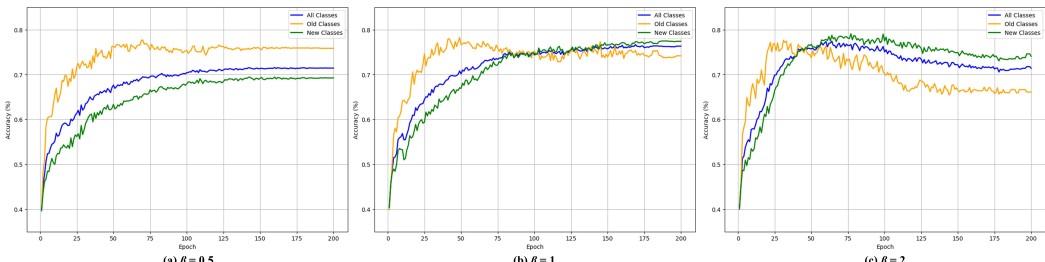

(a) $\beta = 0.5$   (b) $\beta = 1$   (c) $\beta = 2$

Figure 5: Visualization of the influence of the ETF loss coefficient $\beta$ on CUB-200.

Our method consistently achieves the best performance across all backbones. With DINOv1 and DINOv2 as backbones, NC-GCD slightly outperforms SelEx by up to +1.5% in Old accuracy and +1.0% in New accuracy, demonstrating stable and balanced gains. It should be noted that SelEx, GCD, and SimGCD did not originally adopt CLIP as their backbone—we thus reproduced these methods with CLIP as the backbone. When using the CLIP backbone, our method's improvements become more significant, achieving +4.3% in All accuracy and +9.8% in Old accuracy. This result indicates that our consistent supervised–unsupervised alignment can effectively leverage semantic priors from large-scale vision–language pretraining.

Averaged over all backbones, NC-GCD reaches 80.4%, 80.5%, and 80.3% on All, Old, and New categories, respectively, outperforming all existing methods. These results verify that the proposed framework maintains strong backbone-agnostic consistency: by fixing Equiangular Tight Frame (ETF) prototypes, NC-GCD enforces a unified geometric structure regardless of the feature distribution of the encoder. Moreover, the large gain under CLIP shows that our geometric alignment is complementary to semantic alignment, resulting in a more structured and discriminative representation space for both known and novel categories. NC-GCD exhibits robust generalization across diverse backbones, demonstrating its universality and adaptability for generalized category discovery tasks.

### B.3 Analysis of ETF Loss Coefficient $\beta$

The ETF loss coefficient $\beta$ plays a crucial role in balancing the contributions of the ETF alignment loss and the representation learning loss in our model. As shown in Table 9 and Fig. 5, varying $\beta$ significantly influences both the "Best Accuracy" (Best ACC) and "Final Accuracy" (Final ACC)

Table 10: Impact of estimation bias in category number $K$ on CUB-200 and CIFAR-100.

| Bias in $K$ | CUB-200 | | | CIFAR-100 | | |
|---|---|---|---|---|---|---|
| | All (%) | Old (%) | Novel (%) | All (%) | Old (%) | Novel (%) |
| -20% | 70.3 | 72.1 | 69.4 | 79.8 | 83.7 | 72.5 |
| -10% | 72.5 | 73.5 | 72.0 | 81.2 | 84.3 | 75.2 |
| 0% | **74.8** | **76.8** | **73.8** | **82.7** | 85.5 | **77.3** |
| +10% | 74.2 | 75.2 | 73.7 | 82.6 | **85.7** | 76.8 |
| +20% | 73.5 | 74.5 | 73.0 | 81.4 | 84.9 | 74.8 |

during training. "Best Accuracy" (Best ACC) refers to the highest accuracy achieved by the model at any point during the training process, typically before the model converges. It represents the model's peak performance in classifying both known and novel categories during training. "Final Accuracy" (Final ACC) refers to the accuracy achieved at the last epoch, which indicates the model's final performance after all epochs. This measure is typically considered the most reliable indicator of how well the model generalizes to unseen data.

In Fig. 5, we observe the accuracy curves across various values of $\beta$. When $\beta = 1$ (Fig. 5b), the model demonstrates the best overall performance with the highest Best ACC of 76.67% and a solid Final ACC of 76.33%, indicating an optimal balance between ETF alignment and representation learning. The model shows excellent performance in both known and novel categories, with consistent improvements across all category types. The curves in Fig. 6b stabilize, reflecting the model's ability to learn stable features across both known and new categories without overfitting.

When $\beta$ is reduced to 0.5 (Fig. 5a), we see a noticeable decrease in both Best ACC (71.57%) and Final ACC (71.44%), especially for novel categories, where the accuracy drops to 69.44%. The model's performance on novel categories is notably weaker compared to the scenario when $\beta = 1$, as reflected in the flatter curves for new categories in Fig. 6a. The lower $\beta$ value reduces the contribution of ETF alignment, impairing the model's ability to effectively separate categories geometrically, which results in poorer category discovery and less effective feature alignment.

On the other hand, increasing $\beta$ to 2 (Fig. 5c) leads to a higher Best ACC for novel categories (78.75%) but a significant drop in Final ACC for old categories, which falls to 66.11%. While the model achieves better performance on novel categories during early training, it eventually overfits to the new categories, as seen in the accuracy curves for new categories. This causes a performance imbalance, with the model neglecting the stability of learned representations for known categories. As a result, the Final ACC for old categories is significantly lower, leading to an overall drop in model performance.

In conclusion, the results show that $\beta = 1$ strikes the optimal balance, achieving the highest and most stable performance across both known and novel categories. The Best ACC indicates the model's peak performance, while the Final ACC reflects its robustness in the final phase of learning. As our evaluation focuses on Final ACC, the results indicate that $\beta = 1$ is the best choice for stable and high performance in generalized category discovery tasks, ensuring effective feature alignment and balanced learning across categories.

### B.4 Impact of Incorrect Estimation of Category Number $K$

In the NC-GCD framework, the pre-assigned ETF prototypes rely on the estimated number of categories $K$ to construct the Simplex ETF structure. However, accurately determining the true number of novel categories is often difficult in real-world scenarios. To investigate the model's robustness to estimation errors, we systematically evaluate NC-GCD under different levels of estimation bias, specifically with deviations of $\pm 10\%$ and $\pm 20\%$ from the ground-truth value, using the CUB-200 and CIFAR-100 datasets.

As shown in Table 10, underestimating $K$ leads to a significant decline in performance, particularly for novel categories. When $K$ is underestimated by 20%, the novel category accuracy on CUB-200 drops to 69.4%, and All category Accuracy (All ACC) decreases to 70.3%. In contrast, overestimating $K$ by 20% results in a much smaller drop, with All ACC still maintaining 73.5%. A similar trend is observed on CIFAR-100, where underestimating $K$ by 20% causes the novel category accuracy to

Table 11: Comparison of category number estimation ($K$) and all category accuracy (All ACC) across different methods.

| Method | CIFAR100 | | ImageNet100 | | CUB | | Stanford Cars | | FGVC Aircraft | | Herbarium19 | |
|---|---|---|---|---|---|---|---|---|---|---|---|---|
| | $K$ | All ACC | $K$ | All ACC | $K$ | All ACC | $K$ | All ACC | $K$ | All ACC | $K$ | All ACC |
| Ground Truth | 100 | - | 100 | - | 200 | - | 196 | - | 100 | - | 683 | - |
| GCD [4] | **100** | 70.8 | 109 | 77.9 | 230 | 51.1 | 230 | 39.1 | - | - | 520 | 37.2 |
| GPC [12] | 100 | 75.4 | 103 | 75.3 | **201** | 52.0 | **201** | 38.6 | - | - | - | - |
| PIM [7] | 95 | 75.6 | **102** | 83.0 | 227 | 62.0 | 169 | 55.1 | - | - | 563 | 42.0 |
| CMS [8] | 97 | 79.6 | 116 | 81.3 | 170 | 64.4 | 156 | 51.7 | **98** | 55.2 | **666** | 37.4 |
| **Ours (NC-GCD)** | 96 | **80.5** | 109 | **85.7** | 182 | **70.3** | 169 | **54.0** | 105 | **55.4** | 568 | **42.3** |

fall to 72.5% and All ACC to 79.8%, while overestimating $K$ by 20% still preserves relatively high novel accuracy of 74.8% and All ACC of 81.4%.

These results confirm that underestimating $K$ has a more detrimental impact on novel category discovery due to insufficient prototype diversity, whereas the model is more tolerant to overestimation. This observation supports our conservative $K$ estimation strategy, where slightly overestimating the number of categories is preferred to mitigate the negative effects of prototype under-representation.

Overall, these findings demonstrate that accurate estimation of $K$ is crucial for optimal performance, particularly in novel category discovery. Underestimating $K$ significantly hampers the model's ability to generalize to novel categories, while overestimating $K$ introduces redundancy without severe performance degradation. This conclusion further validates our conservative estimation strategy adopted in Section B.5, where a slightly larger $K$ is preferred to ensure sufficient prototype capacity. In future work, we will explore adaptive prototype adjustment techniques to dynamically refine $K$ during training and further enhance model robustness in open-world settings.

## B.5 Performance under Estimated Category Number $K$

In real-world scenarios, the exact number of novel categories is often unknown. Instead of assuming access to the ground-truth $K$, our NC-GCD framework is capable of estimating the number of categories directly from data through clustering-based methods. This section evaluates how well the model performs under such automatic estimation compared to other state-of-the-art approaches.

We conduct an initial CMS-style [8] representation learning phase for 30 epochs and estimate the number of categories every 5 epochs based on the current feature space. Following our earlier analysis on the impact of ETF prototype number deviations, we adopt a conservative estimation strategy by selecting the maximum estimated $K$ across all estimation points. This approach effectively reduces the negative impact of underestimating $K$, which, as previously shown, leads to significant performance degradation on novel categories, while the model remains more tolerant to overestimation.

As shown in Table 11, our NC-GCD achieves the highest All category Accuracy (All ACC) across all evaluated datasets, despite moderate estimation biases in $K$. On challenging fine-grained datasets such as CUB and Stanford Cars, our method significantly outperforms other baselines. Specifically, NC-GCD achieves 70.3% All ACC on CUB, surpassing CMS by +5.9%, and 54.0% on Stanford Cars, improving over CMS by +2.3%. On Herbarium19, even with a slight underestimation of $K$, our method still leads by a considerable margin, achieving 42.3% All ACC compared to the next best 37.4% by CMS.

For more generic datasets like CIFAR100 and ImageNet100, where the estimated $K$ values are closer to the ground truth, NC-GCD continues to deliver the best performance, reaching 80.5% and 85.7% All ACC, respectively. These results demonstrate that our method is not only effective in fine-grained scenarios but also robust across diverse open-world classification tasks.

Although the estimated $K$ may deviate from the true value, especially in complex fine-grained settings, NC-GCD maintains superior accuracy and shows strong resilience to estimation errors. In future work, we plan to explore advanced model selection and adaptive prototype adjustment techniques to further improve the estimation accuracy of $K$ and enhance the overall performance of generalized category discovery.

Table 13: Comparison of parameters and accuracy.

| Method | Total parameter | Trainable parameter | Avg Acc |
|---|---|---|---|
| SimGCD (ICCV 2023) | 92.89M | 20.74M | 62.57 |
| CMS (CVPR 2024) | **92.49M** | 20.34M | 64.08 |
| **NC-GCD (Ours)** | 92.89M | **20.34M** | **68.70** |

### B.6 Analysis of Clustering Frequency $T$

We further analyze the sensitivity of our NC-GCD framework to the clustering frequency, controlled by the parameter $T$, which determines how often clustering and prototype updates occur. As shown in Table 12, setting $T = 5$ achieves the best overall performance, with the highest **All** accuracy of **74.80%** and the highest **Novel** accuracy of **73.8%**. This indicates that performing clustering every five epochs strikes an effective balance between feature stability and timely prototype updates, leading to improved novel category discovery.

Interestingly, when $T = 10$, the accuracy for **Old** categories peaks at **77.5%**, suggesting that less frequent clustering better preserves the learned representations of known categories. However, this comes at the cost of reduced **Novel** accuracy, highlighting a trade-off between stability for known categories and adaptability for novel categories.

When the clustering frequency is too high ($T = 1$), the model suffers from unstable prototype updates, negatively affecting both **Old** and **Novel** accuracies. Conversely, setting $T = 20$ significantly reduces the model's ability to adapt to novel categories due to infrequent updates, leading to the lowest overall performance.

Table 12: Impact of $T$ on CUB-200.

| Clustering Period $T$ | Accuracy | | |
|---|---|---|---|
| | All | Old | Novel |
| 1 | 73.7 | 75.2 | 72.9 |
| 5 | **74.8** | 76.8 | **73.8** |
| 10 | 74.2 | **77.5** | 72.6 |
| 20 | 72.9 | 74.5 | 71.9 |

## C Parameter and SCM Efficiency Analysis

We analyze the computational efficiency of our method from two perspectives: (1) model parameter efficiency and (2) the computational cost introduced by the Semantic Consistency Matcher (SCM).

### C.1 Parameter Analysis

We analyze the computational efficiency of our method by comparing the total number of parameters, the trainable parameters, and the average accuracy (Avg ACC) achieved by our approach, SimGCD, and CMS.

As shown in Table 13, our method achieves competitive performance with an average accuracy of **68.7%**, significantly outperforming SimGCD and CMS, which achieve accuracies of 62.57% and 64.08%, respectively. Our method has a similar total parameter count (92.89M) compared to SimGCD (92.89M), but unlike SimGCD, where the classifier is trained alongside the rest of the model, we pre-assign ETF prototypes and do not require training a classifier. This results in the same number of trainable parameters (20.34M) as CMS, which does not use a classifier either.

This comparison demonstrates that our approach does not introduce additional trainable parameters compared to methods like CMS, which do not rely on classifiers but still achieve significantly higher accuracy. The efficiency of our model is evident in the fact that it improves performance without requiring extra learnable parameters for the classifier, as seen in the results. In conclusion, our method achieves high performance with highly efficient parameterization, making it an attractive solution for generalized category discovery tasks, as it strikes a balance between computational cost and accuracy.

Table 14: Computation time comparison (per iteration) on CUB-200 (3090 GPU).

| Method | Feature Extraction (s) | Clustering (s) | SCM Matching (ms) | Class Center Computation (s) |
|---|---|---|---|---|
| CMS (CVPR 2024) | 17.2846 | 4.2050 | – | – |
| **NC-GCD (Ours)** | 17.2846 | 4.2050 | **0.138** | **0.2828** |

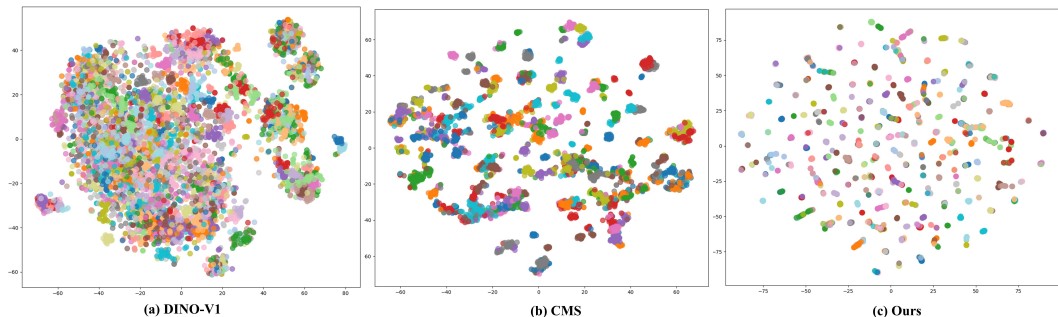

(a) DINO-V1      (b) CMS      (c) Ours

Figure 6: Visualization of all categories of CUB-200.

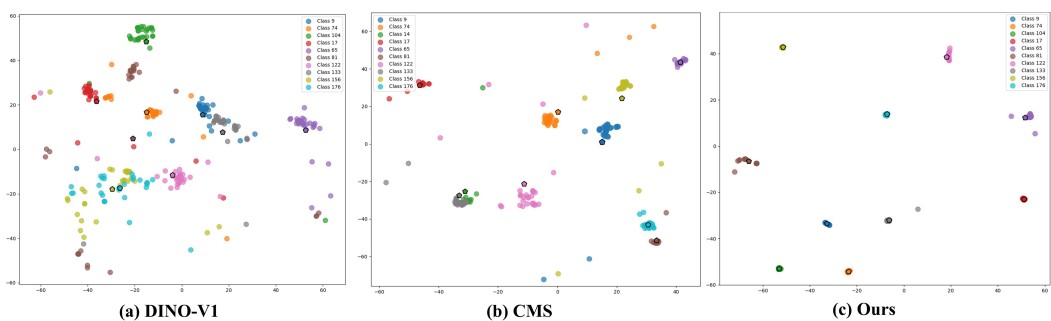

(a) DINO-V1      (b) CMS      (c) Ours

Figure 7: Visualization of 10 random categories of CUB-200.

## C.2 SCM Efficiency

The Semantic Consistency Matcher (SCM) effectively improves the stability of clustering assignments while introducing minimal computational overhead. As reported in Table 14, SCM Matching takes only **0.138 ms** per execution, and the additional time for category center computation is just **0.2828 s**. Moreover, SCM is executed only once every **5 epochs**, which further reduces its impact on the overall training time.

All experiments are conducted on a single NVIDIA 3090 GPU, and even under this setting, the additional overhead introduced by SCM remains negligible.

Despite this minimal overhead, SCM plays a critical role in improving label consistency and enhancing model performance. Notably, our NC-GCD framework achieves **significant performance gains with almost no additional computational cost**, demonstrating an excellent trade-off between accuracy and efficiency. These results highlight the practical scalability and real-world applicability of NC-GCD for generalized category discovery tasks.

## D    More Visualization Results

### D.1    Comparison with State of the Art Methods

To further demonstrate the effectiveness of our NC-GCD framework, we present additional visualizations of the feature space learned by our method. The t-SNE visualizations of all categories and 10 random categories from the CUB-200 dataset are shown in Figures 6 and 7, respectively.

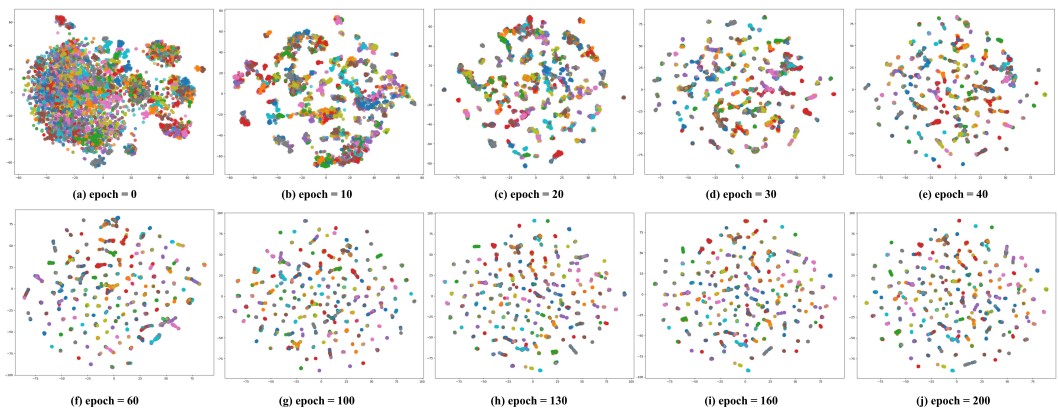

Figure 8: Visualization of our training process on CUB-200 (All categories).

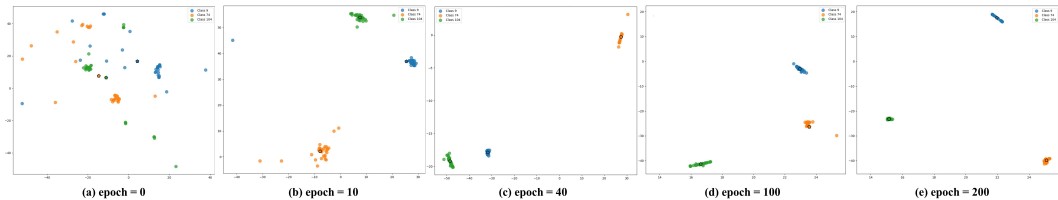

Figure 9: Visualization of our training process on CUB-200 (3 random categories).

In Fig. 6, we visualize the embeddings of all categories in the CUB-200 dataset. Fig. 6 (a) shows the t-SNE visualization of the embeddings extracted using DINOv1, where significant overlap between categories is observed. This overlap indicates that the model struggles to distinguish between many of the categories. Fig. 6 (b) presents the embeddings extracted by CMS, which also show notable clusters but still suffer from some confusion and overlap. In contrast, Fig. 6 (c) shows the embeddings from our NC-GCD method, where the categories are more clearly separated. This demonstrates our model's ability to learn more distinct and structured feature representations for all categories, confirming the efficacy of our pre-assigned ETF prototypes and alignment strategy in improving class separability.

Fig. 7 provides a zoomed-in view with t-SNE visualizations of 10 randomly selected categories from the CUB-200 dataset. Fig. 7 (a) again shows the embeddings from DINOv1, where the categories are still widely scattered and not well separated. Fig. 7 (b) shows CMS embeddings, where the clustering has improved but still lacks the fine-grained separation seen in (c). In Fig. 7 (c), our method successfully groups the categories with minimal overlap, and the embeddings show compact, well-separated clusters. This highlights our method's ability to preserve class separability even with a limited subset of categories, further demonstrating the robustness of NC-GCD in handling both known and novel categories.

These visualizations underscore the power of our NC-GCD framework in ensuring clear category separation and reducing category confusion. The effectiveness of the ETF alignment strategy is evident in both the overall category separability and the detailed random category embeddings, where our method significantly outperforms both DINOv1 and CMS in organizing the feature space.

## D.2 Training Process

To better understand how our NC-GCD framework organizes feature representations over time, we provide t-SNE visualizations of the training process on the CUB-200 dataset. These visualizations demonstrate how our method progressively refines feature distributions, ultimately achieving a well-structured and highly separable feature space.

Fig. 8 illustrates the evolution of feature representations for all categories across different training epochs. At epoch 0, the features are highly entangled, with significant category confusion due to the lack of geometric constraints. As training progresses, the feature clusters gradually become more

distinct, aligning with their respective ETF prototypes. By epoch 40, the initial separation between categories emerges, and by epoch 100, the majority of categories are well-structured. By epoch 200, the features exhibit a near-complete collapse to a simplex ETF structure, aligning closely with the Neural Collapse (NC) theory [11], where features within each class concentrate around their mean while inter-class distances remain maximized.

Fig. 9 further highlights this phenomenon by showing three randomly selected categories. Initially, these categories exhibit substantial overlap, but as training progresses, their feature distributions become increasingly compact while maintaining distinct separation. Eventually, each category collapses onto a single point, which is consistent with the expected NC behavior. This confirms that our method successfully enforces a geometric structure that maximizes category separability while maintaining within-category compactness.

These visualizations validate the effectiveness of our approach. By leveraging Neural Collapse principles and ETF alignment, NC-GCD ensures a consistent and optimal feature arrangement throughout training. The final collapse of features to a well-defined simplex ETF structure highlights the theoretical soundness of our method and its strong alignment with NC theory, making it highly effective for Generalized Category Discovery.

## E    Limitations and Future Work

While our NC-GCD framework demonstrates strong performance across various benchmarks, there remain opportunities for further improvement.

First, the scalability of the fixed ETF prototype structure has not been thoroughly explored. Although the current design works well on the evaluated datasets, its effectiveness in large-scale or highly complex scenarios remains to be validated. Future work will investigate how to extend the ETF framework to better handle diverse and large-scale category distributions.

Second, the estimation of the category number $K$ relies on clustering-based heuristics, which generally perform well but may be sensitive to feature quality in some challenging scenarios. Incorporating more robust estimation techniques could help improve performance stability, especially in cases with highly ambiguous data distributions.

## F    Broader Impacts

This work explores a novel framework for Generalized Category Discovery (GCD), enhancing the ability of AI systems to autonomously recognize and differentiate new concepts in open-world environments. By reducing the dependence on large-scale labeled datasets, our method lowers the barriers to applying AI technologies in domains where data annotation is expensive or difficult to obtain[4, 48].

The proposed framework encourages more efficient use of unlabeled data, offering practical solutions for knowledge discovery in fields such as environmental monitoring, healthcare diagnostics, and industrial quality control. Furthermore, it has the potential to accelerate the development of adaptive intelligent systems capable of continuously learning from new, unseen data without extensive human supervision. We hope this research will contribute to broader advancements in learning with limited supervision and inspire further innovations in building more flexible and scalable AI systems[49].

