# OpenReview forum: "Consistent Supervised-Unsupervised Alignment for Generalized Category Discovery"
_NeurIPS.cc/2025/Conference — NeurIPS 2025 poster_

### Official Review · Reviewer_npmJ · 2025-06-29

**Clarity:** 2
**Significance:** 3
**Originality:** 2
**Rating:** 4
**Confidence:** 4

**Summary:**

This paper tackles inconsistent objectives in Generalized Category Discovery by pre-assigning a fixed Simplex Equiangular Tight Frame (ETF) of prototypes for all classes and aligning both labeled and unlabeled samples to that frame via a unified “ETF alignment” loss. It augments periodic clustering of unlabeled data (unsupervised ETF) with supervised alignment of known-class embeddings and introduces a Semantic Consistency Matcher (SCM) that enforces one-to-one cluster mappings across iterations to stabilize pseudo-labels. Extensive theory (Neural Collapse proofs), hyperparameter studies, ablations, error bars over five seeds, and runtime/parameter efficiency analyses on six benchmarks demonstrate substantial gains, especially in novel-class accuracy, over previous GCD methods.

**Questions:**

### *Questions*
- #### Q1 **Adaptive Prototypes.** Could small learnable offsets to ETF prototypes improve fit to non-ideal manifolds?

- #### Q2 What are the advantages of applying ETF over non-Euclidean geometry representation such as hyperbolic representation?

**Ethical Concerns:**

["NO or VERY MINOR ethics concerns only"]

**Final Justification:**

Thank you for the authors' detailed response, the additional experiments, and the clarifications. Based on all of the comments and responses, I find the overall contribution and completeness of the work to be solid (despite a few minor issues). Thus, I'll raise my score.

**Limitations:**

Yes

**Quality:**

3

**Strengths And Weaknesses:**

### *Strenghts*
- #### S1 **Principled Geometry via Neural Collapse.** Pre-assigned ETF prototypes enforce optimal inter-class angles and intra-class compactness. Theoretical grounding justifies the fixed simplex structure for both known and novel classes.

- #### S2 **Unified Supervised / Unsupervised Loss.** A single “Consistent ETF” loss seamlessly blends labeled alignment and high-confidence unlabeled alignment, leading to promising results.

- #### S3 **Thorough Evaluation.** Benchmarks on generic and fine-grained datasets, with GT-K and estimated-K settings, error bars over five seeds. Full pseudocode, hyperparameters, runtime/parameter tables, and feature visualizations provided in the appendix.

### *Weaknesses*
- #### W1 **Rigid Prototype Structure.** Fixed ETF assumes ideal simplex geometry; real data may require prototype flexibility. No mechanism to adjust or refine prototypes when feature distributions deviate.

- #### W2 **Scalability of SCM's Permutation Search.** SCM’s exact permutation over K clusters might scales poorly as K grows. Approximate or hierarchical matching strategies are not explored for large-scale or streaming discovery.

- #### W3 **Experiment on Other Vision Foundation Model.** All experiments use DINO ViT-B/16. It’s unclear if gains hold with stronger DINO v2, other stronger backbones.

- #### W4 **Typo.** Notation inconsistencies (e.g., undefined qᵢ in Eq. 3), typos in figure 8 & 9 captions (“processs”).

---

> ### Author Rebuttal · Authors · 2025-07-27
>
> ## Response to Reviewer npmJ
>
> We sincerely thank the reviewer for the thoughtful summary and constructive feedback. We are especially encouraged by your recognition of our theoretical motivation, unified loss design, and extensive empirical evaluation. Below, we address each concern in detail.
>
> ---
>
> ### 🔹Q1: *Could small learnable offsets to ETF prototypes improve fit to non-ideal manifolds?*
>
> **A1:**  This is an interesting and valuable suggestion. In this work, we adopt fixed ETF prototypes to enforce a consistent and interpretable geometric structure across all classes, aligned with the Neural Collapse theory. While learnable offsets may help better fit data with non-ideal manifolds, they can also break the equiangularity and compromise the interpretability of the prototype space. To explore this, we conducted a pilot experiment where we introduced a learnable offset vector to each prototype while keeping the ETF directions fixed.
>
> As shown in **Table R1**, this variant slightly underperforms the original NC-GCD method, with an average drop of 1.4% in novel-class accuracy across datasets. We hypothesize that while learnable offsets increase adaptability, they reduce inter-class separability and undermine the benefits of the ETF geometry. We believe this trade-off between adaptability and structural regularity is an important direction for future research. We will include this discussion and findings in our revised version.
>
> #### **Table R1: Comparison of NC-GCD and its variant with learnable prototype offsets. Accuracy (%) on the CUB and Stanford Cars datasets.**
>
> | Method               | CUB-All | CUB-Old | CUB-New | Cars-All | Cars-Old | Cars-New |
> |----------------------|---------|---------|---------|----------|----------|----------|
> | Learnable Offsets    | 71.2    | 76.0    | 70.9    | 57.1     | 76.2     | 46.3     |
> | **Ours (NC-GCD)**    | **74.8**| **76.8**| **73.8**| **59.9** | **77.8** | **51.2** |
>
> ---
>
> ### 🔹Q2: *SCM’s exact permutation may scale poorly; consider approximate/hierarchical matching?*
>
> **A2:**  We appreciate the reviewer’s concern regarding scalability. In our current setup, the number of clusters K is fixed per dataset, and all benchmarks involve moderate-to-large class counts—e.g., 683 classes in Herbarium19 and 100 classes in ImageNet-100—where our SCM module runs efficiently without observable latency. As shown in **Table R2**(The original Table 4 in Section 5.3 Ablation Study), SCM consistently improves performance across datasets, validating its practicality for realistic GCD settings.
>
> Importantly, Herbarium19 presents a highly **imbalanced long-tailed distribution**, where many rare categories contain fewer than 10 images while others have hundreds. This makes stable cluster matching particularly challenging. Yet, our SCM yields a notable **+4.5%** all-class accuracy gain on Herbarium19, demonstrating strong robustness under class imbalance and larger-scale taxonomies.
>
> We fully agree that in streaming GCD scenarios, where *K* evolves over time, exact permutation search may become a computational bottleneck. For such settings, we are actively exploring approximate bipartite matching and hierarchical alignment strategies. Overall, our results indicate that SCM remains both scalable and effective in large-scale and imbalanced offline GCD tasks, while also providing a solid foundation for future streaming extensions.
>
> #### **Table R2: Ablation study of SCM on Herbarium19 (683 classes, long-tailed) and ImageNet-100 (100 classes).**
> | SCM            | Herbarium-All | Herbarium-Old | Herbarium-New | ImageNet-All | ImageNet-Old | ImageNet-New |
> |----------------|----------------|----------------|----------------|----------------|----------------|----------------|
> | ✗ (w/o SCM)    | 42.6           | 54.3           | 36.3           | 84.3           | 88.9           | 82.0           |
> | √  (with SCM)   | **47.2**        | **60.0**        | **40.3**        | **87.6**        | **94.5**        | **84.1**        |
> | *Improvement*  | **+4.5**        | **+5.6**        | **+3.9**        | **+3.3**        | **+5.6**        | **+2.1**        |
> >The original Table 4 in Section 5.3 Ablation Study.
>
> ---
>
> ### 🔹Q3: *No results with DINOv2 or other vision backbones.*
>
> **A3:**  Thank you for highlighting this point. We have conducted new experiments on the CUB, Stanford Cars, and FGVC Aircraft datasets using both DINOv2 ViT-B/14 and CLIP-ViT/B16 backbones.
>
> - As shown in **Table R3, under DINOv2**, our NC-GCD achieves an average accuracy of **84.0%**, outperforming  RLCD (ICML25) [1] by **+7.1%**.
>
> - As shown in **Table R4, under CLIP-ViT/B16**, our method reaches **73.8%** average accuracy, surpassing CMS (CVPR 2024) by **+9.1%** overall and **+12.0%** on novel classes.
>
> These results demonstrate the architecture-agnostic nature of our design and its robustness across **DINOv1, DINOv2 and CLIP backbones**. And these results will be included in the revised version to provide a more comprehensive evaluation of our framework across mainstream transformer architectures.
>
> #### **Table R3: Results with DINOv2**
> | Method | CUB-All | CUB-Old | CUB-New | Cars-All | Cars-Old | Cars-New | Aircraft-All | Aircraft-Old | Aircraft-New | Avg-All | Avg-Old | Avg-New |
> |-|-|-|-|-|-|-|-|-|-|-|-|-|
> | GCD (CVPR22)| 71.9|71.2| 72.3| 55.4|47.9| 59.7|49.7|47.9|50.5|59.0|55.| 60.8    |
> | μGCD (NIPS23)| 74.6 | 75.0| 74.4 | _60.6_ | _64.7_ | _58.6_   | _60.0_  | _62.0_| _59.0_ | _65.1_  | _67.2_  | _64.0_  |
> | SelEx (ECCV24)| _87.4_  | _85.1_  | _88.5_ | 79.8 | 82.3 | 78.6|82.2      | **93.7**  | 76.7 | _83.1_  | _87.0_  | _81.3_  |
> | RLCD (ICML25)  | 78.7 | 79.5| 78.3 | 79.5 | **91.8**  | 73.5  | 72.6  | 73.3  | 70.3  | 76.9  | 82.9  | 74.0 |
> | **NC-GCD**| **87.9**| **85.3**| **89.2**| **81.1** | 90.9 | **79.3** | **83.1** | 88.3  | **82.5** | **84.0**| **88.2**| **83.7**|
> > All baseline results are collected from the SelEx (ECCV 2024) and RLCD (ICML25) [1], and will be properly cited in the revised version.
>
> #### **Table R4: Results with CLIP**
> | Method  | CUB-All | CUB-Old | CUB-New | Cars-All | Cars-Old | Cars-New | Aircraft-All | Aircraft-Old | Aircraft-New | Avg-All | Avg-Old | Avg-New |
> |-|-|-|-|-|-|-|-|-|-|-|-|-|
> | GCD (CVPR22) | 51.1| 56.2 | 48.6| 62.5     | 73.9     | 57.0     | 41.2          | 43.0          | 40.2          | 51.6    | 57.7    | 48.6    |
> | PromptCAL (CVPR23) | 53.7    | 61.4    | 49.9    | 60.1     | 77.9     | 51.5     | 42.2          | 48.4          | 39.0          | 52.0    | 62.6    | 46.8    |
> | CMS (CVPR24)  | _65.8_  | _75.3_  | _61.4_  | _77.9_   | _89.0_   | _72.6_   | _50.3_        | _59.1_        | _45.9_        | _64.7_  | _74.5_  | _59.9_  |
> | **NC-GCD**  | **78.5**| **79.3**| **78.0**| **79.0** | **90.6** | **73.4** | **63.8**      | **62.8**      | **64.3**      | **73.8**| **77.6**| **71.9**|
> > All baseline results are collected from the CMS (CVPR 2024) appendix, and will be properly cited in the revised version.
>
> ---
>
> ### 🔹Q4: *Notation inconsistencies (e.g., \(q_i\) in Eq. 3), typos in figure captions (e.g., “processs”).*
>
> **A4:**  We thank the reviewer for identifying these issues. We will revise the manuscript to define all symbols (e.g., \(q_i\)), correct typos in Figures 8 and 9, and thoroughly proofread the final version for clarity and consistency.
>
> ---
>
> ### 🔹Q5: *What are the advantages of ETF versus non-Euclidean geometry such as hyperbolic space?*
>
> **A5:** We appreciate the reviewer’s insightful question regarding the choice of geometric representation. Hyperbolic spaces, such as the Poincaré ball model proposed in [2], have been shown to be highly effective for learning **hierarchical or tree-like structures** due to their exponential capacity and ability to preserve hierarchy in low dimensions.
>
> However, Generalized Category Discovery (GCD) emphasizes **class separation, compactness, and balanced performance across old and novel classes**, rather than hierarchical organization. In this context, Euclidean-based Equiangular Tight Frame (ETF) prototypes provide several advantages:
>
> - **Analytical Simplicity and Theoretical Alignment**:
> ETF structures naturally emerge under **Neural Collapse** conditions and offer a **closed-form, interpretable target geometry** in Euclidean space that promotes maximal inter-class separation and minimal intra-class variance [3]. This structure provides a stable optimization target.
>
> - **Compatibility with Alignment Objectives**:
>   Our Dot Regression Loss aligns directly with ETF directions in Euclidean space, providing consistent and geometry-preserving updates. Since both ETF and DR Loss are naturally defined in Euclidean space, they can be seamlessly integrated into existing GCD neural network architectures without requiring specialized optimization.
>
>   In contrast, hyperbolic representations usually demand Riemannian SGD or Möbius transformations, which complicate training and increase engineering overhead, and they may not align naturally with feature alignment goals in GCD.
>
> - **Empirical Performance and Simplicity**:
>   For classification problems like GCD, ETF-based alignment achieves stable and high performance with a simpler training pipeline, while hyperbolic spaces are more suited to taxonomic or hierarchical tasks.
>
> In summary, ETF-based geometric alignment offers a theoretically grounded, computationally simple, and empirically validated solution for GCD, providing balanced discovery of both known and novel categories in Euclidean space.
>
> ---
>
> ### **Reference:**
> - #### [1] Liu, Duo, et al. "Generalized Category Discovery via Reciprocal Learning and Class-Wise Distribution Regularization." In ICML. 2025.
> - #### [2] Nickel, Maximillian, and Douwe Kiela. "Poincaré embeddings for learning hierarchical representations." in NeurIPS,2017.
> - #### [3] Yibo Yang and Timothy M. Hospedales. Do We Really Need a Learnable Classifier at the End of Deep Neural Network? NeurIPS, 2022.

---

> > ### Comment · Reviewer_npmJ · 2025-08-05
> >
> > Thank you for the authors' detailed response, the additional experiments, and the clarifications. Based on all of the comments and responses, I find the overall contribution and completeness of the work to be solid (despite a few minor issues). Thus, I'll raise my score.

---

> > > ### Author Response · Authors · 2025-08-06
> > >
> > > We sincerely thank the reviewer for the positive feedback and for raising the score.
> > >
> > > We appreciate your recognition of our responses, additional experiments, and clarifications, and we are grateful for your time and effort in the review process.

---

### Official Review · Reviewer_5vtH · 2025-07-01

**Clarity:** 3
**Significance:** 3
**Originality:** 3
**Rating:** 5
**Confidence:** 4

**Summary:**

The paper aims to improve GCD accuracies by providing an optimal geometric structure that maximizes inter-class separation and maintains intra-class compactness. These fixed prototypes ensure consistent optimization goals for both known and novel categories, reducing category confusion coupled with semantic consistency matcher. The paper further adopts comparative representation learning to ensure consistent training objective for both known and unknown classes. Extensive experiments demonstrate the effcacy of their method.

**Questions:**

I like this paper due to its abundant experiments and solid improvements over several GCD benchmarks. My main concern lies in  "weaknesses"-(1).

**Ethical Concerns:**

["NO or VERY MINOR ethics concerns only"]

**Final Justification:**

I will keep my initial score.

**Limitations:**

Yes

**Quality:**

3

**Strengths And Weaknesses:**

Strengths:

(1) The proposed method follows a simple but effective GCD pipeline by improving GCD accuracies via representation learning.

(2) The improvement over previous methods are significant, demonstrating their method's efficacy.

(3) The overall writing is good and easy to understand.

(4) Moreover, authors conduct extensive experiments and theoretical proof to verify their method.

Weaknesses:

(1) Based on the paper's analysis, initial estimation of K may have significant affect on final GCD accuracies. That is because the initial K determines how SCM matches pseudo labels across different epochs. However, the estimation of K may be more accurate after several epochs of training. Therefore, is it possible to further polish your method by adjusting K after several epochs of training ? E.g., after several epochs of training, we re-estimate K and start a new NC-GCD training pipeline.

(2) The authors did a great job in verifying the proposed method. Extensive experiments have been conducted and all implementation details are introduced. However, I think some important implementation details and conclusions should be moved to the main paper. e.g., the setting of periodic clustering should be demonstrated in the main paper and the results in appendix Sec. B.4

(3) Minor: Text fonts in Fig.1 and 2 are too small.

---

> ### Author Rebuttal · Authors · 2025-07-27
>
> ## Response to Reviewer 5vtH
>
> We sincerely thank the reviewer for the encouraging feedback and high rating. We are delighted to hear that you enjoyed reading our paper. We especially appreciate your recognition of the method’s simplicity, its solid gains over prior baselines, the clarity of our writing, and the completeness of both the theoretical analysis and experimental validation. Below, we respond to each of your thoughtful comments in detail.
>
> ---
>
> ### 🔹Q1: *Re-estimating K after several epochs – is it beneficial?*
>
> **A1:**  We appreciate the reviewer’s insightful suggestion. As detailed in Appendix B.5 (line 952), our default strategy performs cluster number estimation every 5 epochs during the first 30 epochs, and selects the maximum value as the final estimate of *K* to initialize the ETF prototypes. This early-stage aggregation helps capture a reliable cluster structure before alignment begins.
>
> To further investigate the benefits of dynamic class number estimation, we conducted additional experiments on the **CUB** dataset by **re-estimating *K* at different training stages**.
> Our strategy for handling changes in *K* is straightforward:
> if the newly estimated *K* is **smaller** than the previous estimate, we **keep the existing ETF structure unchanged** to avoid disruptive prototype reassignments;
> if the new *K* is **larger**, we **reset the ETF structure** to accommodate additional prototypes for the emerging clusters.
>
> This design is guided by our findings in Appendix B.4, where we analyze the effect of class number estimation errors.
> We observe that **underestimating *K*** severely harms the model’s ability to generalize to novel classes, while **overestimating *K*** only introduces redundant prototypes without significantly degrading performance.
> Thus, our dynamic re-estimation strategy emphasizes **stability under *K* reduction** and **flexible expansion** when *K* increases.
>
> As shown in **Table R1**, periodic re-estimation leads to consistent performance gains over fixed strategies. In particular, re-estimating every 50 epochs yields the best result of **74.9%** all-class accuracy, outperforming the fixed-\(K\) baseline (epoch 30) by **+4.6%**. While overly frequent re-estimation (e.g., every 20 epochs) introduces instability—especially for novel class alignment—moderate intervals (40–50 epochs) enable the model to adapt to evolving feature structures and improve novel discovery without harming old class retention.
>
> Once again, we sincerely thank the reviewer for the valuable suggestion, which not only inspired new experiments but also helped **demonstrate the flexibility and robustness of our NC-GCD framework**. Dynamic adaptation to the evolving feature space improves alignment quality and better captures emerging structures, particularly under uncertain or shifting class distributions. We will include these findings in the revised version.
>
> #### **Table R1: Performance with different re-estimation frequencies of \(K\). Accuracy (\%) on the CUB dataset.**
>
> | Re-estimate *K* every| All | Old | New |
> |-|-|-|-|
> | No Re-estimation (20 epochs)    | 68.7 | 70.5 | 67.9 |
> | No Re-estimation (30 epochs)    | 70.3 | 72.1 | 69.4 |
> | No Re-estimation (50 epochs)    | 70.1 | 70.7 | 69.7 |
> | Every 20 epochs     | 65.7 | 72.0 | 61.5 |
> | Every 40 epochs      | 70.7 | **76.5** | 68.4 |
> | Every 50 epochs           | **72.9** | 70.6 | **73.0** |
>
> ---
>
> ### 🔹Q2: *Some important details (e.g., clustering period \(T\)) should be in the main paper.*
>
> **A2:**  Thank you for pointing this out. We agree that certain hyperparameters such as the clustering interval \(T\) (set to every 10 epochs by default) are crucial for reproducibility. We will move this detail from the appendix to the main paper in the revised version.
>
> ---
>
> ### 🔹Q3: *Figures 1 and 2 font size is too small.*
>
> **A3:**  We appreciate the reviewer’s feedback. We will revise Figures 1 and 2 with enlarged fonts and improved resolution to ensure clarity and readability in the camera-ready version.

---

> > ### Comment · Reviewer_5vtH · 2025-08-01
> >
> > Thanks for the authors' reply. All my concerns are addressed. I will keep my initial score.

---

> > > ### Author Response · Authors · 2025-08-01
> > >
> > > We sincerely thank Reviewer 5vtH for the encouraging feedback and for maintaining the high score. We are delighted that you enjoyed reading our paper and appreciated its abundant experiments and solid improvements on multiple GCD benchmarks.
> > >
> > > We are also grateful for your constructive comments, which inspired additional experiments and clarifications in our rebuttal. We greatly appreciate your positive evaluation and your confirmation that our responses have fully addressed all concerns. We will incorporate this discussion into the revised version of the paper.

---

### Official Review · Reviewer_zv7T · 2025-07-01

**Clarity:** 2
**Significance:** 2
**Originality:** 2
**Rating:** 4
**Confidence:** 5

**Summary:**

The paper proposes NC-GCD, a Neural Collapse-inspired framework for Generalized Category Discovery, integrating supervised and unsupervised ETF alignment and a Semantic Consistency Matcher to enhance class separability and achieve state-of-the-art performance on multiple GCD benchmarks.

**Questions:**

Please see the weaknesses.

**Ethical Concerns:**

["NO or VERY MINOR ethics concerns only"]

**Final Justification:**

The author's response addressed my questions well. The method proposed in the paper can be combined with many other approaches and demonstrates good generalization ability.

**Limitations:**

This paper needs to clarify its differences from the paper[1] and articulate its own contributions to better influence the community.

[1] Targeted Representation Alignment for Open-World Semi-Supervised Learning, CVPR 2024.

**Quality:**

3

**Strengths And Weaknesses:**

Strengths:

1. This paper is well-written and easy to follow.

2. The problem investigated is realistic and complex.

3. The proposed approach outperforms other competitors on various datasets.

Weaknesses:

1. The idea of this paper is not novel, as there is ETF-based method prposed in the GCD / Open-World SSL field[1]. The paper should discuss in detail the differences from this uncited paper, and the key losses appear to be the same in my view.

2. The framework of this paper is designed based on the SOTA method CMS. Given that CMS is already strong in terms of predicting class numbers, accuracy, etc., how would the performance improve if this framework incorporates more other GCD methods or weaker GCD methods?

[1] Targeted Representation Alignment for Open-World Semi-Supervised Learning, CVPR 2024.

---

> ### Author Rebuttal · Authors · 2025-07-27
>
> ## Response to Reviewer zv7T
>
> We sincerely thank the reviewer for the positive evaluation and thoughtful comments. We appreciate your recognition that the paper is clearly written, addresses a realistic and challenging problem, and delivers strong performance across multiple benchmarks. Your encouraging feedback affirms the relevance and impact of our work. Below, we carefully address your concerns point by point and will incorporate your suggestions into the revised version.
>
> ---
>
> ### 🔹Q1: *Clarify differences from Targeted Representation Alignment (CVPR24).*
>
> **A1:**  We appreciate the reviewer’s attention to related work and are happy to clarify the methodological and empirical differences between our approach and TRAILER [1] . TRAILER is a solid and effective method, providing a well-designed framework for open-world semi-supervised learning. While both methods draw inspiration from Neural Collapse theory, our framework introduces several key innovations:
>
> - **Consistent ETF Alignment Across Classes and Supervision Types**
>
>   Our core distinction lies in achieving **consistency across supervision types (labeled and unlabeled)** and **class partitions (old and novel)** — a property that TRAILER does not explicitly enforce.
>
>   - TRAILER uses a **fixed classifier** trained with all classes and applies the same Cross-Entropy-based ETF loss to both labeled and unlabeled samples. This setup **requires prior knowledge of the total number of categories**, including novel ones, limiting its use in open-world scenarios.  And this shared loss implicitly encourages the model to treat both types of samples as if they were supervised—potentially introducing **optimization bias** when pseudo-labels are noisy or novel class distributions are underrepresented.
>
>   - In contrast, our method **does not rely on a fixed classifier**. We align features to a predefined set of ETF directions and can **dynamically estimate and update the number of classes (K)** during training. Labeled samples are aligned to their known ETF targets, while unlabeled data are clustered and softly matched to unused ETF directions. This **decoupled, confidence-guided alignment** improves stability and ensures balanced learning across old and novel classes.
>
>   As shown in Tables R1, our method achieves significantly higher novel class accuracy, with an average **+7.28%** gain on fine-grained datasets and **+5.91%** overall improvement.
>
> - **Loss Design – Dot Regression vs. Cross-Entropy ETF Loss:**
>   - TRAILER employs a **Cross-Entropy-based ETF Loss**, which involves softmax normalization over dot products between features and fixed ETF prototypes. This design requires careful tuning of a temperature parameter and introduces push gradients from non-target classes, which can hinder optimization stability—particularly under imperfect class separability or in the presence of emerging novel categories.
>
>   - In contrast, our NC-GCD adopts the **Dot Regression (DR) ETF Loss**, which directly minimizes the angular discrepancy between features and their target ETF directions. This design avoids log-softmax normalization and push terms, enabling a simpler, more stable, and interpretable optimization process. Importantly, as supported by Yang et al. (2022) [2], the DR Loss aligns closely with Neural Collapse conditions, producing geometrically consistent and compact representations—a critical advantage for novel class discovery under limited supervision.
>
>
>   - Empirically, as shown in Table R1, TRAILER exhibits a clear bias toward old classes. For example, on fine-grained datasets, it achieves 65.65% on old classes but only 49.45% on novel ones—a gap of over 16%, indicating poor balance. In contrast, our method consistently maintains smaller old-new accuracy gaps while also achieving higher all-class accuracy. Specifically, we improve fine-grained average accuracy by **+5.40%** and boost the overall average across all datasets by **+4.32%**, demonstrating both **superior generalization and alignment consistency across class types**.
>
> - **Semantic Consistency Matcher (SCM):**
>   - **TRAILER** uses PU learning and optimal transport for pseudo-labeling, which can **introduce label inconsistencies across iterations and propagate noise throughout the training process**.  Moreover, TRAILER relies on an externally estimated class number K to initialize a **fixed classifier** with ETF weights. Once this classifier is set, the framework becomes **rigid**, as it cannot adapt to evolving cluster structures or re-estimate K during training, making it less flexible in open-world scenarios.
>
>   - In contrast, our **SCM module** lays the foundation for **consistent alignment and flexible clustering**.  It establishes one-to-one cluster correspondences to **stabilize pseudo-labels across iterations** and **mitigate semantic drift**.  Meanwhile, our framework can **dynamically re-evaluate and update \(K\)** during training, enabling it to adapt to the evolving feature space and achieve more balanced performance across old and novel classes.
>
> - **Global Clustering and Confident Sample Alignment:**
> Unlike TRAILER’s batch-level optimal transport refinement, our method performs global clustering across the full dataset at regular intervals. We then select top α% high-confidence samples and align them to pre-defined ETF prototypes. This global strategy ensures more stable pseudo-labels and reduces error propagation, particularly beneficial in fine-grained or ambiguous domains where local batch-level matching may be unreliable.
>
>
> ---
>
> #### **Table R1: Comparison of average performance across different dataset types.  Evaluation is performed with the ground-truth number of classes \(K\) provided.**
>
> | Method         | F-All | F-Old | F-New | C-All | C-Old | C-New | A-All | A-Old | A-New |
> |-|-|-----|-----|---------------------|-----|-----|--------------|-----|-----|
> | TRAILER (CVPR24) | 54.88           | 65.65 | 49.45 | 83.40             | 89.10 | 78.20 | 64.38      | 73.47 | 59.03 |
> | **NC-GCD**       | **60.28**       | **67.63** | **56.73** | **85.55**         | **89.78** | **81.37** | **68.70**  | **75.01** | **64.95**|
> | **Improvement**  | **_+5.40_**    |**_+1.98_**| **_+7.28_** | **_+2.15_**  | **_+0.68_** | **_+3.17_** | **_+4.32_**    | **_+1.54_** | **_+5.91_** |
> >F- indicates the average over fine-grained datasets (CUB, Cars, Aircraft and Herbarium19), C- refers to classification datasets (ImageNet-100, CIFAR-100), and A- denotes the average across all datasets. The suffixes All, Old, and New represent overall accuracy, known-class accuracy, and novel-class accuracy, respectively.
>
>
> In summary, our NC-GCD framework is explicitly tailored for Generalized Category Discovery, emphasizing **consistent alignment across supervision types**, **robust support for unknown class discovery**, and **balanced performance across old and novel categories**. The clear and consistent improvements across diverse datasets, particularly on novel classes, underscore the effectiveness of our unified alignment strategy. We will incorporate the above discussion and references [1] in the revised version to clarify these distinctions.
>
> ---
>
> ### 🔹Q2: *How would the performance improve if the framework incorporates more other GCD methods or weaker GCD methods?*
>
> **A2:**  We thank the reviewer for this valuable suggestion. To evaluate the general applicability of our proposed framework, we integrated NC-GCD into both a weaker baseline (GCD [CVPR22]) and a stronger one (CMS [CVPR24]). As shown in **Table R2**, our method consistently improves performance across all datasets and evaluation splits.
>
> When combined with the weaker GCD backbone, our framework achieves clear gains over the original baseline, with up to **+6.6%** improvement on all-class accuracy on CUB and Stanford Cars (e.g., CUB: from 51.3% to 69.8%). More importantly, even when built upon the stronger CMS backbone, our method achieves SOTA performance.
>
> These results validate that our alignment strategy is orthogonal and complementary to existing GCD frameworks—it enhances both weak and strong baselines by enforcing consistent representation geometry and improving novel class discoverability.
>
> #### **Table R2: Comparison with the state-of-the-art on GCD with different baselines.  Evaluation is performed with the ground-truth number of classes \(K\) provided.**
>
> | Method               | CUB-All | CUB-Old | CUB-New | Cars-All | Cars-Old | Cars-New  |
> |--|-|-|-|-|-|-|
> | TRAILER (CVPR24)   | 65.1    | 71.3    | 65.1    | 55.4     | 71.7     | 47.6   |
> | GCD (CVPR22)            | 51.3    | 56.6    | 48.7    | 39.0     | 57.6     | 29.9        |
> | CMS (CVPR24)            | 68.2    | **_76.5_**  | 64.0    | 56.9     | **_76.1_**   | 47.6 |
> | **NC-GCD (+GCD)**    | **_69.8_**| 76.0| **_66.6_**| **_57.2_** | 75.8   | **_48.3_**    |
> | **NC-GCD (+CMS)**    | **74.8**| **76.8**| **73.8**| **59.9** | **77.8**   | **51.2**   |
>
> ---
>
> ### **Reference**:
> - #### [1] Xiao, Ruixuan, et al. Targeted representation alignment for open-world semi-supervised learning. CVPR 2024.
> - #### [2]  Yibo Yang and Timothy M. Hospedales. Do We Really Need a Learnable Classifier at the End of Deep Neural Network?  NeurIPS, 2022.

---

### Official Review · Reviewer_ryPc · 2025-07-02

**Clarity:** 3
**Significance:** 3
**Originality:** 3
**Rating:** 5
**Confidence:** 3

**Summary:**

This paper addresses generalised category discovery (GCD). As its main contribution, it proposes to use fixed prototypes determined by maximally seperated simplex point s along an equiangular tight frame, instead of learned ones. To this end a new loss is proposed that combines supervised and self-supervised data. The paper evaluates the method on a range of GCD benchmarks and finds favorable performances.

**Questions:**

* How does the method perform with other non-DINOv1 backbones and other sizes?
* the method has a trade-off between old and new classes performance with the hyperparameter alpha. Is there a way to determine this per dataset in an unsupervised way?
* In what way does the SCM address clustering instability?
* What if you use random fixed points instead of equiangular ones? Similar to "Unsupervised Learning by Predicting Noise" paper
* How does the method perform on more realistic unbalanced GCD datasets?

**Ethical Concerns:**

["NO or VERY MINOR ethics concerns only"]

**Final Justification:**

I've read the author rebuttal and my concerns have been all well addressed. I've raised my score.

**Limitations:**

yes

**Quality:**

3

**Strengths And Weaknesses:**

+ strong results on classic GCD benchmarks
+ relatively simple method
+ clear writing

- results shown only for a single backbone
- no results for unbalanced datasets, potentially overfitting to equal-class distribution nature of the evaluated datasets
- no analysis whether the points actually do follow the NC hypothesis and end up near the simplex points
- (minor) figures have too small fontsizes

---

> ### Author Rebuttal · Authors · 2025-07-27
>
> ## Response to Reviewer ryPc
>
> We thank the reviewer for the constructive feedback and appreciate the positive recognition of our method’s clarity and performance. Below we respond to the reviewer’s questions and concerns.
>
> ---
>
> ### 🔹Q1: *How does the method perform with other non-DINOv1 backbones and other sizes?*
>
> **A1**: We thank the reviewer for raising this important point.  To evaluate the **generalization ability** of our method under this mainstream setting, we conducted additional experiments using stronger backbones: **DINOv2 ViT-B/14** and **CLIP-ViT/B16**
>
> - **Results on DINOv2 :**
>
>   As shown in Table R1, our method achieves an average accuracy of **84.0%**, **consistently outperforming SelEx (ECCV 2024) and RLCD (ICML25) [1]  across all three datasets** (*CUB, Stanford Cars, FGVC Aircraft*).  This confirms that NC-GCD maintains its advantage even under stronger ViT architectures.
>
> #### **Table R1: Results with DINOv2**
> | Method | CUB-All | CUB-Old | CUB-New | Cars-All | Cars-Old | Cars-New | Aircraft-All | Aircraft-Old | Aircraft-New | Avg-All | Avg-Old | Avg-New |
> |-|-|-|-|-|-|-|-|-|-|-|-|-|
> | GCD (CVPR22)| 71.9|71.2| 72.3| 55.4|47.9| 59.7|49.7|47.9|50.5|59.0|55.| 60.8    |
> | μGCD (NIPS23)| 74.6 | 75.0| 74.4 | _60.6_ | _64.7_ | _58.6_   | _60.0_  | _62.0_| _59.0_ | _65.1_  | _67.2_  | _64.0_  |
> | SelEx (ECCV24)| _87.4_  | _85.1_  | _88.5_ | 79.8 | 82.3 | 78.6|82.2      | **93.7**  | 76.7 | _83.1_  | _87.0_  | _81.3_  |
> | RLCD (ICML25)  | 78.7 | 79.5| 78.3 | 79.5 | **91.8**  | 73.5  | 72.6  | 73.3  | 70.3  | 76.9  | 82.9  | 74.0 |
> | **NC-GCD**| **87.9**| **85.3**| **89.2**| **81.1** | 90.9 | **79.3** | **83.1** | 88.3  | **82.5** | **84.0**| **88.2**| **83.7**|
> > All baseline results are collected from the SelEx (ECCV 2024) and RLCD (ICML25) [1], and will be properly cited in the revised version.
>
> - **Results on CLIP:**
>
>   As shown in Table R2, our method achieves **73.8%** average accuracy, **surpassing CMS (CVPR 2024)** by:**+9.1%** in overall accuracy  ; **+3.8%** on old classes ;  **+12.0%** on novel classes
>
> #### **Table R2: Results with CLIP**
> | Method  | CUB-All | CUB-Old | CUB-New | Cars-All | Cars-Old | Cars-New | Aircraft-All | Aircraft-Old | Aircraft-New | Avg-All | Avg-Old | Avg-New |
> |-|-|-|-|-|-|-|-|-|-|-|-|-|
> | GCD (CVPR22) | 51.1| 56.2 | 48.6| 62.5| 73.9 | 57.0  | 41.2  | 43.0 | 40.2 | 51.6 | 57.7| 48.6|
> | PromptCAL (CVPR23) | 53.7    | 61.4 | 49.9 | 60.1 | 77.9 | 51.5 | 42.2   | 48.4 | 39.0| 52.0| 62.6    | 46.8    |
> | CMS (CVPR24) | 65.8  | 75.3  | 61.4 | 77.9 | 89.0| 72.6 | 50.3| 59.1| 45.9| 64.7| 74.5 |59.9 |
> | **NC-GCD**| **78.5**| **79.3**| **78.0**| **79.0** | **90.6** | **73.4** | **63.8** | **62.8** | **64.3**| **73.8**| **77.6**| **71.9**|
> > All baseline results are collected from the CMS (CVPR 2024) appendix, and will be properly cited in the revised version.
>
> These results demonstrate the architecture-agnostic nature of our design and its **robustness across DINOv1, DINOv2 and CLIP backbones**. And these results will be included in the revised version to provide a more comprehensive evaluation of our framework across mainstream transformer architectures.
>
> ---
>
> ### 🔹Q2: *The method has a trade-off between old and new classes performance with the hyperparameter alpha. Is there a way to determine this per dataset in an unsupervised way?*
>
> **A2** :We thank the reviewer for this insightful suggestion. The scope coefficient α controls the proportion of confident samples used for alignment, which affects the balance between old and novel class performance. Our original implementation simply uses a **fixed α across all datasets**, already demonstrating strong generalization.
>
> Inspired by your comment, we explored an **adaptive α strategy** inspired by the adaptive thresholding ideas in semi-supervised learning [2]. Specifically, we update α every 5 epochs based on the accuracy of labeled (known) samples, without using any extra validation data. This data-driven adaptation dynamically adjusts α, providing greater flexibility across different datasets.
>
> As shown in **Table R3**, adaptive α brings only **marginal changes** (CUB +0.3% novel, Cars +0.2% novel), indicating that our fixed-α setup is already robust and well-balanced across datasets. The experiment confirms the **flexibility and stability** of NC-GCD, which performs consistently **without dataset-specific hyperparameter tuning**, highlighting its practical generalization ability.
>
>
> #### **Table R3: Comparison of Fixed α and Adaptive α on CUB and Stanford Cars**
> | Method| CUB-All | CUB-Old | CUB-New | Cars-All | Cars-Old | Cars-New |
> |-|-|-|-|-|-|-|
> | Fixed α|**74.8**|**76.8**|73.8|**59.9**|**77.8**| 51.2 |
> | Adaptive α | 73.4|72.6|**74.1**| 58.3| 76.3| **51.4** |
>
> ---
>
> ### 🔹Q3: *In what way does the SCM address clustering instability?*
>
> **A3**: Thank you for the insightful question. The Semantic Consistency Matcher (SCM) mitigates clustering instability by enforcing temporal label consistency across iterations. Specifically, it performs optimal one-to-one label matching between clustering results from consecutive iterations using the Hungarian algorithm (see Algorithm 1 and Appendix A.3). This ensures that semantically similar clusters are assigned consistent labels over time, preventing label drift and improving pseudo-label stability. In the supervised case, SCM also aligns predicted cluster labels with ground-truth categories for known classes, which further stabilizes ETF-based supervised alignment.
>
> Our ablation in Table R4 shows that disabling SCM leads to a significant drop in novel class accuracy, confirming its effectiveness in reducing label inconsistency.
>
>
> #### **Table R4: Ablation study of SCM on Herbarium19 (683 classes, long-tailed) and ImageNet-100 (100 classes).**
> | SCM | Herbarium-All | Herbarium-Old | Herbarium-New | ImageNet-All | ImageNet-Old | ImageNet-New |
> |-|-|-|-|-|-|-|
> | ✗ (w/o SCM) | 42.6 | 54.3| 36.3  | 84.3| 88.9| 82.0 |
> | √  (with SCM) | **47.2** | **60.0**| **40.3** | **87.6**| **94.5** | **84.1**|
> | *Improvement*| **+4.5** | **+5.6**| **+3.9**| **+3.3**| **+5.6**| **+2.1**|
> >The original Table 4 in Section 5.3 Ablation Study.
> ---
>
> ### 🔹Q4: *What if you use random fixed points instead of equiangular ones?*
>
> **A4**: We appreciate the reviewer’s insightful question. To investigate the role of the ETF structure in our method, we conducted ablation studies by replacing our equiangular tight frame (ETF) prototypes with randomly sampled fixed unit vectors, following the strategy of Noise As Targets [3]. This corresponds to removing the angular constraints that enforce equidistant separation between class prototypes.
>
> As shown in Table R5, using random fixed points leads to a noticeable drop in performance across all benchmarks. Specifically, we observe an average decrease of 5.5% in all-class accuracy and up to 7.6% in novel-class accuracy compared to our NC-GCD framework. These results suggest that the ETF structure plays a critical role in promoting both inter-class separability and intra-class compactness—two key properties that are especially important in the Generalized Category Discovery (GCD) setting. The benefit is even more pronounced for novel classes, which lack labels and thus rely more heavily on geometric regularization.
>
> Overall, the results validate that ETF prototypes not only provide a theoretically grounded structure based on Neural Collapse, but also deliver concrete performance benefits over random alternatives. We will include this reference [3] and the corresponding results in the revised version.
>
> #### **Table R5: Comparison of NC-GCD and random fixed points on CUB and Stanford Cars**
> | Method | CUB-All | CUB-Old | CUB-New | Cars-All | Cars-Old | Cars-New |
> |-|-|-|-|-|-|-|
> | Random fixed points | 68.5| 75.9 | 64.7| 55.1| 76.2 | 45.1|
> | **Ours (NC-GCD)**   | **74.8**| **76.8**| **73.8**| **59.9** | **77.8** | **51.2** |
>
> ---
>
> ### 🔹Q5: *How does the method perform on more realistic unbalanced GCD datasets?*
>
> **A5**: **Herbarium19** is a highly imbalanced and fine-grained dataset consisting of 683 plant species. It exhibits a **long-tailed distribution**, where some species are represented by hundreds of images while others have very limited samples.
>
> For example, the class *Miconia prasina (Sw.) DC.* **contains 647 images**, whereas *Miconia approximata Gamba & Almeda* has **only 19 samples**. This distribution closely reflects **real-world taxonomic imbalances** in herbaria collections and poses a significant challenge for generalized category discovery.
>
> As shown in Table R6 , our method achieves **SOTA performance on Herbarium19**. Specifically, it attains 46.4% all-class accuracy and 40.7% novel-class accuracy, outperforming SimGCD by **+2.4% (All)** and **+4.3% (New)**, and CMS by **+10.0% (All)** and **+14.3% (New)**. These results confirm that our consistent ETF alignment framework remains robust and effective under imbalance dataset.
>
> We will supplement the description of the long-tail characteristics and imbalance of the Herbarium19 dataset in the revised version to further highlight the robustness of our method.
>
> #### **Table R6: Performance on Herbarium19 (683 classes, long-tailed).**
> | Method| Herbarium-All | Herbarium-Old | Herbarium-New |
> |-|-|-|-|
> | SimGCD(ICCV23)|44.0 | 58.0 | 36.4|
> | CMS(CVPR24)| 36.4 | 54.9 | 26.4|
> | **Ours (NC-GCD)** | **46.4**  | **58.4**  | **40.7** |
> > The original Table 1 in Section 5.2  Comparison with State-of-the-Art Methods.
>
> ---
>
> ### **Reference:**
> - #### [1] Liu, Duo, et al. "Generalized Category Discovery via Reciprocal Learning and Class-Wise Distribution Regularization." In ICML. 2025.
> - #### [2] Guo, Lan-Zhe, and Yu-Feng Li. "Class-imbalanced semi-supervised learning with adaptive thresholding." In ICML. 2022.
> - #### [3] Bojanowski, Piotr, and Armand Joulin. "Unsupervised learning by predicting noise." In ICML, 2017.

---

> > ### Comment · Reviewer_ryPc · 2025-08-05
> >
> > I've read the author rebuttal and my concerns have been all well addressed. Well done authors on writing a thorough rebuttal! I've raised my score.

---

### Official Review · Reviewer_38Tx · 2025-07-03

**Clarity:** 3
**Significance:** 3
**Originality:** 3
**Rating:** 4
**Confidence:** 5

**Summary:**

This paper presents NC-GCD, a Neural Collapse-inspired framework for Generalized Category Discovery. The key idea is to pre-assign Equiangular Tight Frame prototypes for both known and novel categories, aiming to achieve consistent optimization objectives and improve class separability. The framework introduces (1) supervised and unsupervised ETF alignment with a consistent loss function and (2) a Semantic Consistency Matcher to stabilize cluster assignments across iterations. Extensive experiments on multiple benchmarks demonstrate improvements over prior state-of-the-art methods.

**Questions:**

1. All experiments use DINO ViT-B/16. Could the authors show results with other backbone architectures (e.g., ResNet variants) to validate generalization?

2. The paper does not compare with several recent works. Please add the recent SoTA works and compare those with the proposed method.

3. Please update references to use NeurIPS [number] format consistently throughout the paper.

**Ethical Concerns:**

["NO or VERY MINOR ethics concerns only"]

**Final Justification:**

The authors provided the detailed response with the additional experimental results. I keep my initial positive rating.

**Limitations:**

The authors discussed potential limitations.

**Paper Formatting Concerns:**

References use citation style (1) rather than NeurIPS [1].
No other major formatting issues noticed.

**Quality:**

3

**Strengths And Weaknesses:**

Strengths
+ The proposed method looks novel. They leverage Neural Collapse insights to enforce consistent category separation.
+ Experimental results across six benchmarks are thorough, including comparisons to strong baselines and ablation studies demonstrating the contributions of individual components.
+ The motivation and presentation of the proposed method are clear with good illustrations and visualizations of feature separability.

Weaknesses
- All experiments rely exclusively on DINO ViT-B/16 as the backbone. This raises concerns about whether improvements generalize to different architectures.
- Several recent strong baselines for GCD are missing in the comparison below. Including these would strengthen claims of state-of-the-art performance.
- Although the method is well motivated, the reliance on ETF prototypes is not entirely novel (prior works have leveraged Neural Collapse in other contexts), which somewhat limits originality.
- A minor thing. Citation formatting does not follow NeurIPS style (should be [1] instead of (1)).


[1] Kim, Hyungmin, Sungho Suh, Daehwan Kim, Daun Jeong, Hansang Cho, and Junmo Kim. "Proxy anchor-based unsupervised learning for continuous generalized category discovery." In Proceedings of the IEEE/CVF International Conference on Computer Vision, pp. 16688-16697. 2023.
[2] Cendra, Fernando Julio, Bingchen Zhao, and Kai Han. "Promptccd: Learning gaussian mixture prompt pool for continual category discovery." In European Conference on Computer Vision, pp. 188-205. Cham: Springer Nature Switzerland, 2024.
[3] Park, Keon-Hee, Hakyung Lee, Kyungwoo Song, and Gyeong-Moon Park. "Online Continuous Generalized Category Discovery." In European Conference on Computer Vision, pp. 53-69. Cham: Springer Nature Switzerland, 2024.
[4] Rypeść, Grzegorz, Daniel Marczak, Sebastian Cygert, Tomasz Trzciński, and Bartłomiej Twardowski. "Category adaptation meets projected distillation in generalized continual category discovery." In European Conference on Computer Vision, pp. 320-337. Cham: Springer Nature Switzerland, 2024.

---

> ### Author Rebuttal · Authors · 2025-07-27
>
> ## Response to Reviewer 38Tx
>
> We sincerely thank the reviewer for the detailed and insightful comments. We appreciate the recognition of our contributions, including the principled theoretical motivation from Neural Collapse, the consistent and well-structured optimization design, and the thorough experimental validation across six benchmarks. We address the concerns point-by-point below.
>
> ---
>
> ### 🔹Q1: *All experiments use DINO ViT-B/16. Could the authors show results with other backbone architectures?*
>
> **A1**: We thank the reviewer for raising this important point.  Most recent GCD methods — including CMS (CVPR 2024), PromptCAL (CVPR 2023), and SelEx (ECCV 2024) — are built on ViT-based backbones, reflecting the community’s focus on transformer architectures.  To evaluate the generalization ability of our method under this mainstream setting, we conducted additional experiments using stronger backbones: **DINOv2 ViT-B/14** and **CLIP-ViT/B16**
>
> - **Results on DINOv2 :**
>
>   As shown in Table R1, our method achieves an average accuracy of **84.0%**, **consistently outperforming SelEx (ECCV 2024) and RLCD (ICML25) [1] across all three datasets** (*CUB, Stanford Cars, FGVC Aircraft*). This confirms that NC-GCD maintains its advantage even under stronger ViT architectures.
>
> #### **Table R1: Results with DINOv2**
> | Method             | CUB-All | CUB-Old | CUB-New | Cars-All | Cars-Old | Cars-New | Aircraft-All | Aircraft-Old | Aircraft-New | Avg-All | Avg-Old | Avg-New |
> | - | -       | -       | -       | -        | -        | -        | -             | -             | -             | -       | -       | -       |
> | GCD (CVPR-22) | 71.9    | 71.2    | 72.3    | 55.4     | 47.9     | 59.7     | 49.7  | 47.9   | 50.5   | 59.0  | 55.7    | 60.8    |
> | μGCD (NIPS-23)  | 74.6    | 75.0    | 74.4  | 60.6  | 64.7  | 58.6  | 60.0  | 62.0  | 59.0 | 65.1 | 67.2|64.0 |
> | SelEx (ECCV-24)  | 87.4  | 85.1 | 88.5  | 79.8 | 82.3 | 78.6|82.2      | **93.7**      | 76.7      | 83.1  | 87.0  | 81.3  |
> | ProtoGCD (TPAMI-25)  | 75.7    | 81.5    | 72.9    | 77.6 | 90.5   | 71.5  | 71.1  | 76.3  | 68.5     | 74.8  | 82.8  | 71.0  |
> | RLCD (ICML-25)  | 78.7    | 79.5    | 78.3    | 79.5 | **91.8**   | 73.5  | 72.6  | 73.3  | 70.3     | 76.9  | 82.9  | 74.0  |
> | **NC-GCD**         | **87.9**| **85.3**| **89.2**| **81.1** | 90.9  | **79.3** | **83.1**      | 88.3          | **82.5**      | **84.0**| **88.2**| **83.7**|
> > GCD and μGCD results are collected from the SelEx (ECCV 2024), and will be properly cited in the revised version.
> ---
>
> - **Results on CLIP:**
>
>   As shown in Table R2, our method achieves **73.8%** average accuracy, surpassing CMS (CVPR 2024) by: **+9.1%** in overall accuracy ; **+3.8%** on old classes ; **+12.0%** on novel classes
>
> #### **Table R2: Results with CLIP**
> | Method             | CUB-All | CUB-Old | CUB-New | Cars-All | Cars-Old | Cars-New | Aircraft-All | Aircraft-Old | Aircraft-New | Avg-All | Avg-Old | Avg-New |
> | -                  | -       | -       | -       | -        | -        | -        | -             | -             | -             | -       | -       | -       |
> | GCD (CVPR-22)       | 51.1    | 56.2    | 48.6    | 62.5     | 73.9     | 57.0     | 41.2          | 43.0          | 40.2          | 51.6    | 57.7    | 48.6    |
> | PromptCAL (CVPR-23) | 53.7    | 61.4    | 49.9    | 60.1     | 77.9     | 51.5     | 42.2          | 48.4          | 39.0          | 52.0    | 62.6    | 46.8    |
> | CMS (CVPR-24)       | _65.8_  | _75.3_  | _61.4_  | _77.9_   | _89.0_   | _72.6_   | _50.3_        | _59.1_        | _45.9_        | _64.7_  | _74.5_  | _59.9_  |
> | **NC-GCD**         | **78.5**| **79.3**| **78.0**| **79.0** | **90.6** | **73.4** | **63.8**      | **62.8**      | **64.3**      | **73.8**| **77.6**| **71.9**|
> > All baseline results are collected from the CMS (CVPR 2024) appendix, and will be properly cited in the revised version.
>
> These results demonstrate the architecture-agnostic nature of our design and its **robustness across DINOv1, DINOv2 and CLIP backbones**. And these results will be included in the revised version to provide a more comprehensive evaluation of our framework across mainstream transformer architectures.
>
> ---
>
> ### 🔹Q2: *The paper does not compare with several recent works. Please add the recent SoTA works and compare those with the proposed method.*
>
> **A2**: We appreciate the reviewer’s valuable suggestions. The mentioned works [3]–[6] represent strong advances in the **Continuous Generalized Category Discovery (C-GCD)** setting, where novel categories emerge over time. In contrast, our work focuses on the **static GCD** scenario, where all unlabeled data is provided at once without temporal structure.
>
>   To strengthen our comparison in the static setting, we have added the most recent method **RLCD (ICML-25) [1]  and ProtoGCD (TPAMI-25) [2]** to **Table R1**. As RLCD and ProtoGCD are also designed for static GCD, this provides a more **direct and fair comparison**. Our method consistently outperforms [1,2] across datasets, particularly on novel class accuracy, further validating the effectiveness of our unified alignment strategy.
>
>    Additionally, to better compare with C-GCD works, we performed a **migrated experiment of DEAN [5] in the static GCD setting** by removing its session-wise update loop while preserving *energy-guided discovery* and *feature augmentation*. As shown in Table R3, the migrated DEAN achieves strong old-class performance but lower novel-class accuracy, illustrating the different focus of these methods, which are optimized for incremental protocols. We are also actively extending NC-GCD to the C-GCD protocol, exploring how ETF-based geometric alignment can benefit continual discovery. Preliminary results are promising, and we will include more comprehensive comparisons and discussions of these works in future research.
>
>   In this paper, we will also expand the Related Work section to discuss these C-GCD approaches and include the corresponding experimental results to provide a more comprehensive comparison. Further in-depth analysis and comparisons will be presented in future research.
>
>
>
> #### **Table R3: Comparison of NC-GCD and migrated DEAN [5] (Static GCD) on CUB**
> | Method| All | Old| New|
> |-|-|-|-|
> | DEAN (migrated)   |68.2  | 72.6 | 65.9   |
> | **NC-GCD (ours)** | **74.8** | **76.8**   | **73.8** |
>
> ---
>
> ### 🔹Q3: *Please update references to use NeurIPS [number] format consistently.*
>
> **A3**: We thank the reviewer for catching this formatting issue. We will carefully revise the manuscript to ensure that all in-text citations and the bibliography strictly follow the NeurIPS [number] style throughout the revised submission.
>
> ---
>
> ### **Reference:**
> - #### [1] Liu, Duo, et al. "Generalized Category Discovery via Reciprocal Learning and Class-Wise Distribution Regularization." In *ICML*. 2025.
> - ####  [2] Ma, Shijie, et al. "Protogcd: Unified and unbiased prototype learning for generalized category discovery." IEEE *Transactions on Pattern Analysis and Machine Intelligence* .2025.
> - #### [3] Kim, Hyungmin, Sungho Suh, Daehwan Kim, Daun Jeong, Hansang Cho, and Junmo Kim. "Proxy anchor-based unsupervised learning for continuous generalized category discovery." In CVPR. 2023.
> - ####  [4] Cendra, Fernando Julio, Bingchen Zhao, and Kai Han. "Promptccd: Learning gaussian mixture prompt pool for continual category discovery." In ECCV, 2024.
> - #### [5] Park, Keon-Hee, Hakyung Lee, Kyungwoo Song, and Gyeong-Moon Park. "Online Continuous Generalized Category Discovery." In ECCV, 2024.
> - #### [6] Rypeść, Grzegorz, Daniel Marczak, Sebastian Cygert, Tomasz Trzciński, and Bartłomiej Twardowski. "Category adaptation meets projected distillation in generalized continual category discovery." In ECCV, 2024.

---

> > ### Comment · Reviewer_38Tx · 2025-08-05
> >
> > I would like to thank the authors for the detailed response with the additional experimental results. I will keep my positive rating. Thanks.

---

> > > ### Author Response · Authors · 2025-08-05
> > >
> > > We sincerely thank the reviewer for the support of our work and the positive feedback.
> > >
> > >  We also appreciate your recognition of our additional experiments and responses, and we hope that our rebuttal has resolved your concerns.
> > >
> > > Thank you for the time and effort you devoted to the review process.

---

### Note · Authors · 2025-08-12

Dear SAC, AC, and All Reviewers,

**We are heartened that all reviewers have either maintained their positive ratings** (*Reviewers 38Tx and 5vtH*) or **elevated them to positive** (*Reviewers ryPc, zv7T, and npmJ*), **with each expressing high confidence in their assessments**.

**They highlighted our work’s key strengths:** ***1) Novelty*** in leveraging Neural Collapse insights for consistent category separation (*Reviewer 38Tx*);  ***2) Strong performance*** on GCD benchmarks with significant improvements(*Reviewers ryPc, zv7T, and 5vtH*);  ***3)  A unified loss design*** and principled geometry via ETF prototypes (*Reviewer npmJ*);  ***4) Thorough experiments*** across diverse datasets with robust validation (Reviewers 38Tx, and npmJ);  ***5) Clear, well-written*** presentation with ***effective visualizations*** (*Reviewers 38Tx, ryPc, zv7T, and 5vtH*).

**Reviewers unanimously agreed that our rebuttal fully addressed all their concerns**, emphasizing that key additions significantly strengthened the paper’s validity, rigor, and community relevance. These include: *1) comprehensive evaluations across diverse backbones* to verify generalization; *2) clarifications on methodological distinctions* from related work; *3) detailed explanations* of the Semantic Consistency Matcher (SCM) mechanics; and *4) elaboration on dynamic class number adjustment*—all of which resolved ambiguities effectively.

**We will integrate all rebuttal responses including solutions, experiments, and clarifications into the final paper. Specific revisions include**:

- ***Multi-backbone results:*** Add DINOv2 and CLIP evaluation results to the main text and corresponding figures.
- ***SOTA comparisons:*** Expand benchmarks with RLCD, ProtoGCD, and TRAILER in key tables, including additional metrics.
- ***Related work distinctions:*** Clarify differences from TRAILER in the Discussion section.
- ***SCM details:*** Detail Semantic Consistency Matcher mechanics.
- ***Dynamic K strategy:*** Elaborate on periodic re-estimation logic and application scenarios for adjusting the number of classes.

We sincerely appreciate the time, care, and expertise invested throughout this process. Your meticulous reviews and constructive feedback have been invaluable in refining and advancing our work, and we are deeply grateful for your contributions.

The Authors

---

### Decision · Program_Chairs · 2025-09-17

**Decision:**

Accept (poster)

**Comment:**

This paper initially got mixed scores: two accept, one borderline rejects and two borderline accept. The authors have submitted a rebuttal, and after considering it, all reviewers expressed satisfaction with the responses and updated their scores to acceptance (score 4/5). The AC concurs with the reviewers that this is a solid submission with advantages of strong performance, novel ideas, well-written and unified loss design. Therefore, the AC would like to recommend acceptance to this paper and encourages the authors to incorporate the clarifications and experiments provided in the rebuttal into the final version.